# Highly concentrated trehalose induces prohealing senescence-like state in fibroblasts via CDKN1A/p21

Jun Muto [1✉], Shinji Fukuda [2], Kenji Watanabe[3], Xiuju Dai[1], Teruko Tsuda[1], Takeshi Kiyoi[4], Kenji Kameda[1], Ryosuke Kawakami[5], Hideki Mori[1], Ken Shiraishi[1], Masamoto Murakami [1], Takeshi Imamura[5,6], Shigeki Higashiyama[7,8], Yasuhiro Fujisawa[1], Yoichi Mizukami[3] & Koji Sayama [1]

Trehalose is the nonreducing disaccharide of glucose, evolutionarily conserved in invertebrates. The living skin equivalent (LSE) is an organotypic coculture containing keratinocytes cultivated on fibroblast-populated dermal substitutes. We demonstrated that human primary fibroblasts treated with highly concentrated trehalose promote significantly extensive spread of the epidermal layer of LSE without any deleterious effects. The RNA-seq analysis of trehalose-treated 2D and 3D fibroblasts at early time points revealed the involvement of the CDKN1A pathway, the knockdown of which significantly suppressed the upregulation of *DPT*, *ANGPT2*, *VEGFA*, *EREG*, and *FGF2*. The trehalose-treated fibroblasts were positive for senescence-associated β-galactosidase. Finally, transplantation of the dermal substitute with trehalose-treated fibroblasts accelerated wound closure and increased capillary formation significantly in the experimental mouse wounds in vivo, which was canceled by the CDKN1A knockdown. These data indicate that high-concentration trehalose can induce the senescence-like state in fibroblasts via CDKN1A/p21, which may be therapeutically useful for optimal wound repair.

[1] Department of Dermatology, Ehime University Graduate School of Medicine, Toon, Japan. [2] Department of Biochemistry, School of Dentistry, Aichi Gakuin University, Nagoya, Japan. [3] Institute of Gene Research, Yamaguchi University Science Research Center, Yamaguchi, Japan. [4] Department of Pharmacology, School of Medicine, Kanazawa Medical University, Uchinada, Japan. [5] Department of Molecular Medicine for Pathogenesis, Ehime University Graduate School of Medicine, Toon, Japan. [6] Translational Research Center, Ehime University Hospital, Toon, Japan. [7] Division of Cell Growth and Tumor Regulation, Proteo-Science Center, Ehime University, Toon, Japan. [8] Department of Molecular and Cellular Biology, Osaka International Cancer Institute, Osaka, Japan. ✉email: muto.jun.xs@ehime-u.ac.jp

Trehalose (α-D-glucopyranosyl α-D-glucopyranoside) is the nonreducing disaccharide of glucose, evolutionarily conserved in eukaryotes, plants, and invertebrates, but does not exist in vertebrates[1]. A pivotal method of synthesis has dramatically reduced the production cost[2]. Trehalose has been demonstrated as multifunctional and was utilized to stabilize lipids, proteins, enzymes, and tissues[2]. Therefore, the organ preservation solution, which was named extracellular-type trehalose-containing solution, was demonstrated to be more effective in preserving lung quality after clinical lung transplantation compared with the primary solution[3]. Interestingly, Tanaka et al. reported that trehalose palliates the polyglutamine-mediated pathology of Huntington's disease in mouse models[4]. Additionally, trehalose inhibits the proliferation of fibroblasts owing to its inhibition of fibroblast transformation into myofibroblasts[5], and induces a reduction in insulin/IGF-1-like signaling, which extends the life span of fibroblasts through an aging-suppressor function[6].

Bioengineered cellularized skin substitutes are frequently used in clinical applications as an alternative grafting technique to autografting and as in vitro study models, although most of the currently available models are epidermal sheets that only repair wounds by inducing keratinocytes rather than guiding a regeneration process. Multi-layered living skin equivalents (LSEs) containing bi-layered constructs that model the epidermal layer with the differentiated keratinocytes on the dermal substitutes cultivated with the fibroblasts had been used to treat skin ulcers of burn injury or epidermolysis bullosa[7]. The fibroblasts in the dermal matrix of LSEs drive epidermal proliferation and differentiation through reciprocal action[8]. Numerous biomedical materials, such as type I collagen, acellular human dermis, collagen-glycosaminoglycan matrices, human plasma, and fibrin glue, have been applied as dermal matrix alternatives. Nevertheless, an ideal matrix that is beneficial, readily available, and has minimal toxicity is yet to be discovered[9]. The effect of highly concentrated trehalose on fibroblasts for LSE development remains elusive. During the course of the trials, we unexpectedly found beneficial effects of trehalose for LSE development.

Cellular senescence has been reported as a stress response in which cells experience stable cell cycle arrest following stress-inducing stimuli. The most conventional senescence marker is senescence-associated β-galactosidase (SA β-gal) activity detected after an increase in lysosomal content[10]. These senescent cells maintain metabolic capabilities and feature a hypersecretory phenotype termed the senescence-associated secretory phenotype (SASP)[11]. Although reports have characterized SASP in various cell types, its detailed composition remains unclear. The SASP is composed of a collection of proinflammatory cytokines, chemokines, and growth factors, such as epiregulin (EREG), FGF2, and VEGF[11]. The senescent cells present transiently during wound repair seemed to be restricted in fibroblast-like cells, which produce platelet-derived growth factor-A (PDGFA) and CCN1 (Cellular Communication Network Factor 1)-enriched SASP to facilitate cutaneous wound healing and limit fibrosis[12,13]. In contrast, fibroblasts in which senescence is induced by oncogenic RAS oversecrete more granulocyte-macrophage colony-stimulating factor (GM-CSF) and IL-6, but not EREG or vascular endothelial growth factor (VEGF), than cells in which senescence is induced by other means such as X-irradiation[11]. Basisty et al. reported the "SASP Atlas," which is a comprehensive proteomic database of soluble proteins originating from multiple senescence inducers and cell types, as well as other candidate biomarkers of cellular senescence that include growth/differentiation factor 15 (GDF15), stanniocalcin 1 (STC1), and SLC1A5[14]. Furthermore, Lamin B1(LMNB1) loss is a robust marker of senescence. There is a decline in *LMNB1* mRNA levels during senescence due to a decrease in *LMNB1* mRNA stability[15].

Cell cycle arrest is another feature of senescent cells that is controlled by the activation of the p53 antiproliferative function. The most pertinent function of p53 in senescence is the acceleration of cyclin-dependent kinase inhibitor 1 A (*CDKN1A*) transcription[16]. The *CDKN1A* gene is a major target of the p53 transcription factor, and its product, p21, is a cyclin-dependent kinase inhibitor, which induces cell cycle arrest[17]. In a p53-induced senescence model, Akt activation and cooperation between p21 and Akt were mandatory for cellular senescence phenotype induction[18]. Polo-like kinase 1 (PLK1) is a key molecule in the G2/M transition. The induction of CDKN1A rapidly decreases cellular levels of the *PLK1* promoter activity[19]. Furthermore, high levels of p21 induce G2 arrest in normal human fibroblasts[20].

In this study, we investigated the effect of trehalose on fibroblasts mixed in type 1 collagen gel and tested whether it could affect LSE construction. We performed RNA-seq of the treated fibroblasts. Subsequently, the therapeutic potential of trehalose-treated fibroblasts in the dermal substitute as a biological dressing was investigated via skin grafting onto the full-thickness wounds of BALB/cAJc1-nu nude mice in vivo. These results provide an avenue for the development of a novel organotypic skin culture system for future therapeutic exploitation.

## Results

**Rapid spread of LSEs containing trehalose in the fibroblast-populated collagen gel**. LSEs have been used to treat skin defects. However, the production of LSEs takes approximately 4 weeks, which makes LSE production impractical for applications in regenerative medicine. To investigate the beneficial effects of trehalose on fibroblasts, we constructed fibroblast-populated type I collagen gel with trehalose, upon which normal human keratinocytes were seeded to form LSEs. The sizes of the LSEs were observed after 2 weeks of airlifting at 37 °C (Fig. 1a). We evaluated the diameters of LSEs prepared in Transwell-COL with a 24-mm insert in a six-well culture plate, and the diameters of LSEs prepared with trehalose were significantly larger than those of LSEs prepared without trehalose after 2 weeks of airlifting (Fig. 1b, c). We confirmed this phenomenon in skin fibroblasts and keratinocytes derived from cells of three other patients 1 week after air exposure (Supplementary Fig. 1a–c).

Two weeks after airlifting, hematoxylin and eosin staining was used to compare the LSEs containing trehalose (10 or 100 mg/ml) with the control LSEs. Interestingly, they were morphologically indistinguishable besides the size of the final products (Fig. 1d). Next, paraffin-embedded sections of LSEs were subjected to immunohistochemistry with the Ki67 antibody to assess the proliferation of fibroblasts. Conversely, fibroblasts in the collagen gel with trehalose showed a significantly increased number of Ki67 positive cells on the dermal side (Fig. 1d and Supplementary Fig. 2). Additionally, we examined elastic and collagen fibers in the three-dimensional culture system with or without trehalose by Elastica van Gieson staining. Histological analysis revealed that collagen fiber (stained red) and elastic fiber (stained black) were morphologically similar among the three groups (Supplementary Fig. 3a) Hyaluronan can be detected histologically by using hyaluronan-binding protein (HABP). Interestingly, the HABP staining did not reveal differences in hyaluronan (HA) distribution between the LSEs with or without trehalose (Supplementary Fig. 3b). Alcian blue staining (pH 2.5) was used to visualize the formation of sulfated and carboxylated acid mucopolysaccharides and sialomucins in the LSEs (stained blue), with no significant changes observed between the three groups (Supplementary Fig. 3c). α-Smooth muscle actin (α-SMA) is used as a marker for myofibroblasts, which is a subset of activated fibrogenic cells. In the LSEs, α-SMA-positive cells were similarly densely lined at the

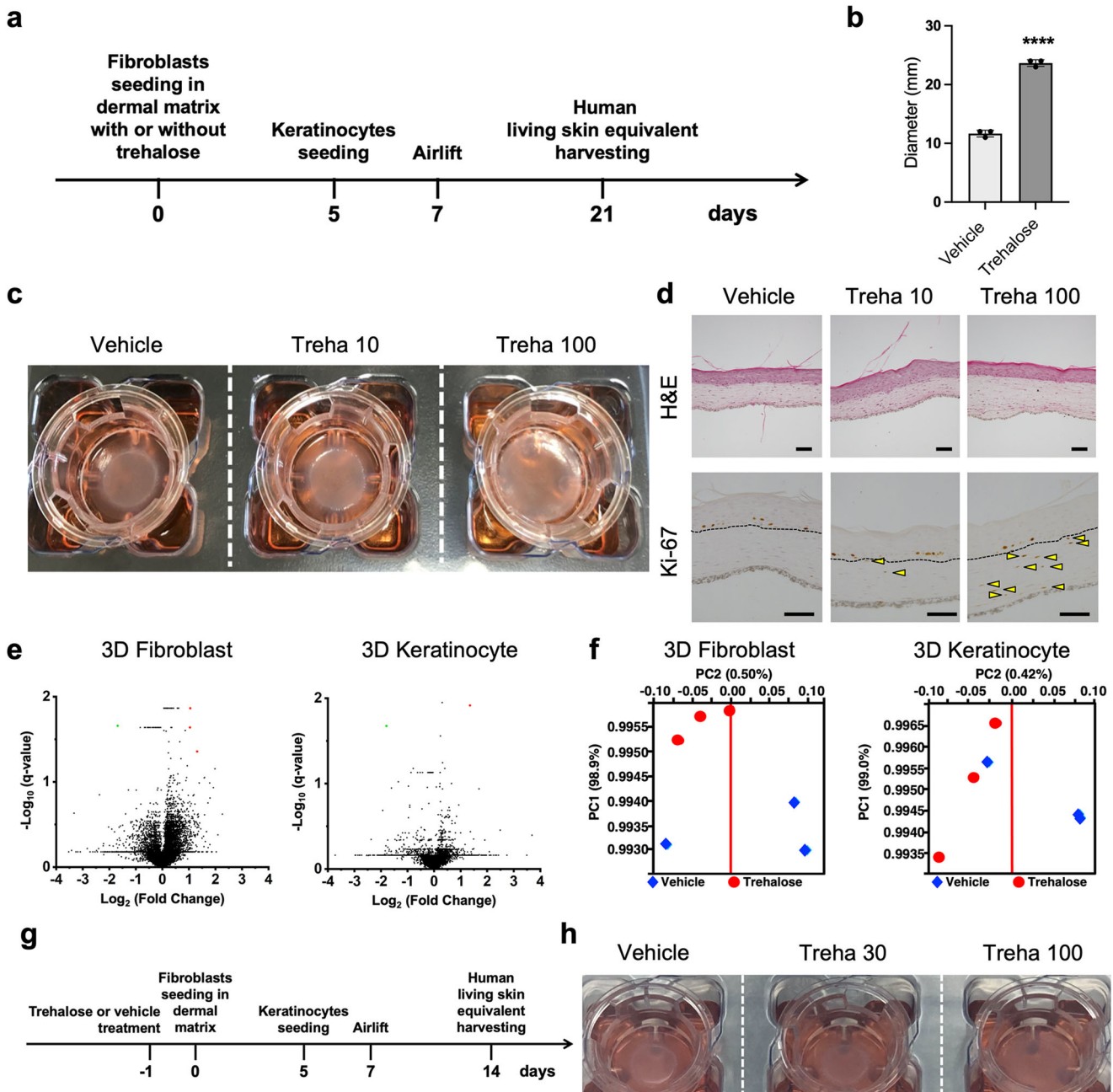

**Fig. 1 Effect of trehalose in the preparation of living skin equivalents. a** Schematic for the preparation of cultured skin equivalents. **b** Diameters of LSEs with or without trehalose (100 mg/ml) added in the collagen gel, prepared in the Transwell-COL with 24-mm insert in a six-well culture plate after 2-week airlifting at 37 °C. Data were expressed as means ± SD for three LSEs, which are representative of three independent experiments with similar results. ****$P < 0.0001$ versus vehicle control groups using the Student t-test. **c** Macroscopic pictures of LSEs with or without trehalose (10 and 100 mg/ml) added in the collagen gel after 2-week airlifting. **d** LSEs stained with hematoxylin and eosin (Scale bar = 50 μm). Paraffin-embedded sections of LSEs were sectioned and subjected to immunohistochemistry with Ki67 antibody. Yellow arrowheads indicate the Ki67 positive fibroblasts in the dermis (Dotted line: dermal–epidermal junction, Scale bar = 100 μm). **e** Volcano plots showing gene expression in the absence or the presence of trehalose. Red or green rounds indicate genes that increased by more than 2-fold or decreased by less than half, respectively, with less than 0.05 of q values. **f** A principal component analysis (PCA) with gene expressions in the absence or the presence of trehalose showed no clear separation between principal component PC1 and PC2. **g** Schematic for the preparation of cultured skin equivalents with fibroblasts treated with or without trehalose before seeding in the dermal matrix. **h** Representative picture of LSEs with the fibroblast treated with or without trehalose (30 and 100 mg/ml) before seeding in the collagen gel after 1-week airlifting at 37 °C. Data were representative of three independent experiments.

dermal–epidermal junction of the three groups (Supplementary Fig. 3d). These data demonstrated that trehalose added to the collagen gel significantly accelerated the proliferation of epidermal sheets, which are morphologically and histologically indistinguishable from vehicle-treated control LSEs.

To investigate the novel effect of trehalose further, we prepared larger LSEs using a larger culture insert (75-mm diameter), with proportionally more fibroblasts and keratinocytes. A rubber ring (8-mm interior diameter) was covered over the fibroblast-containing gel to stabilize it, and keratinocytes were seeded in the

ring hole. The epidermal layer of LSEs containing trehalose (100 mg/ml) in the gel harvested after 2-week airlifting at 37 °C spread markedly, and thus, the experimental LSEs were substantially larger than the control LSEs under the same conditions (Supplementary Fig. 4).

To examine the signaling pathways modulated by trehalose treatment in the 3D skin model, we comprehensively analyzed mRNA expressions in the epidermis and dermis of LSEs cultured in collagen gel containing trehalose (100 mg/ml) 2 weeks after air exposure. RNA-sequencing (RNA-seq) analysis revealed that genes significantly modulated by trehalose were undetected at 14 days in the keratinocytes except for the upregulated bone morphogenic protein 6 (BMP6) gene and the downregulated LINC00302 gene (Fig. 1e). In the fibroblasts, the gene expressions by trehalose resembled those in the vehicle cells, and only four genes (SCARNA22, PTCHD4, RP11-137H2.6, and NPR3) were significantly regulated by the addition of trehalose (Fig. 1e). Principal component analysis (PCA) provided no major difference in the gene expression of the keratinocytes and fibroblasts by trehalose treatment (Fig. 1f). In addition, the factor loadings of genes in PC1 and PC2 showed no effect on trehalose treatment. Our observations indicated that gene expression profiles in a long culture of 3D gels are similar for untreated and trehalose-treated cells in both keratinocytes and fibroblasts. Next, we examined whether fibroblast pretreated with trehalose before seeding in the collagen gel achieved increased proliferation of the epidermal layer of the 3D culture model (Fig. 1g). Interestingly, LSEs with fibroblast pretreated with trehalose (30 or 100 mg/ml) accelerated the spread of the epidermal layer compared with the control LSEs (Fig. 1h). These observations indicated that trehalose pretreatment on the fibroblast monolayer before seeding in the gel can induce significantly accelerated proliferation of the keratinocyte layer of LSE.

**Whole transcriptome analysis in trehalose-treated 2D and 3D fibroblasts**. Gene expression profiles of both keratinocytes and fibroblasts in a long culture of 3D gels treated with trehalose were similar to those of the untreated cells. Thus, we comprehensively examined the transient gene expressions in the trehalose-pretreated fibroblasts using RNA-seq. Trehalose (100 mg/ml) treatment for 24 h induced upregulation of 1256 genes (FC >2.0, $q < 0.01$) and downregulation of 484 genes (FC <0.5, $q < 0.01$) in the 2D culture compared with those of untreated cells. In the 3D culture, 267 genes or 332 genes were upregulated (FC >2, $q < 0.01$) or downregulated (FC <0.5, $q < 0.01$), respectively, by 72 h trehalose treatment (Supplementary Fig. 5a, d). The gene expression profiles of the fibroblasts in 2D and 3D cultures were clearly separated by PC2 in PCA from those of untreated cells, indicating that trehalose affects cellular function through gene expressions (Supplementary Fig. 5b, e). We plotted the factor loadings of the genes in PC1 and PC2 to observe the gene expressions involved in PC2 separation. As the positively contributing genes, growth factors, such as dermapontin (DPT), EREG, FGF2, and angiopoietin-2 (ANGPT2), were observed in both 2D and 3D cultures in addition to a cell cycle inhibitor, CDKN1A (Supplementary Fig. 5c, f). The cell cycle-related genes, Aurora kinase A (AURKA), PLK1, and Myb proto-oncogene like 2 (MYBL2) negatively participated in the PC2 separation of 2D and 3D cultures treated with trehalose (Supplementary Fig. 5c, f). Treatment with highly concentrated trehalose in the fibroblast cells suggests strict regulation of the cell cycle despite the release of various growth factors.

To elucidate the signaling pathways activated in highly concentrated trehalose-treated fibroblasts, we analyzed the interaction network with Ingenuity Pathway Analysis (IPA) using the information collected from databases on protein interactions.

In the presence of trehalose, 131 genes were downregulated in common in 2D and 3D culture, and the expression patterns were shown on the heatmap, which included cell cycle-related genes such as Aurora kinase B (AURKB), PLK1, and Anillin actin-binding protein (ANLN) (Fig. 2a, b). The downregulated genes revealed the reduction of kinetochore metaphase signaling, G2/M DNA damage, and the cell cycle checkpoint in the canonical pathways (Fig. 2c). With respect to upstream factors, asparaginase, a drug for acute lymphoblastic leukemia, was detected in the upstream analysis and is able to arrest the cell cycle. ZBTB17 is a transcriptional negative regulator in the cell cycle (Fig. 2d). CDKN1A was also detected as an upstream factor based on the significant decrease of PLK1, CDK1, CCNA2, and CDC25A after trehalose treatment (Fig. 2e). The inhibition of CDK1 and CCNA2 was suggested to partially induce the reduction of MYBL2. The 127 upregulated genes were observed in both 2D and 3D cultures (Fig. 2f, g), and contained CDKN1A, which was detected as an upstream factor of downregulated genes (Fig. 2d). The pathway analysis using upregulated genes revealed the p53 signaling pathway involving in cell cycle arrest (Fig. 2h). Inhibition of AURK and ANLN were detected as upstream factors in the upregulated genes and were also observed in downregulated genes in the presence of trehalose (Fig. 2i, j). We integrated the upregulated genes into the downregulated genes and analyzed the signaling pathways because the pathways detected by the upregulated genes were closely related to the pathways of the downregulated genes (Fig. 3a). The graphical summary connected the network analysis to the cellular functions and showed that the senescence cells were activated by p53 and CDKN1A related to the cell cycle regulation and mitosis arrest (Fig. 3b). In the network analysis, activation of DPT and VEGF were suggested to be induced by the Notch and Caspase (Supplementary Fig. 6a, b). The mRNA expressions of genes involving cellular senescence, CDKN1A, and LMNB1 were confirmed by quantitative PCR (qPCR) in 2D fibroblasts (Fig. 4a). Western blot analysis revealed that trehalose treatment increased p21 expression and decreased lamin B in a dose-dependent manner (Fig. 4b). The dose-dependently increased expression of p21 in the nuclei after trehalose treatment was confirmed using fluorescence microscopy (Fig. 4c and Supplementary Fig. 7). Furthermore, mRNA expressions of genes involved in cellular functions such as cell cycle regulation, AURKA, AURKB, PLK1, and MYBL2 were confirmed by qPCR in 2D fibroblasts (Fig. 4d) and 3D fibroblasts (Fig. 4e). Additionally, we also confirmed these effects of trehalose with the fibroblasts derived from three other patients. These findings suggested that trehalose possesses the ability to temporarily upregulate CDKN1A and downregulate AURKA, AURKB, UBE2, PLK1, MYBL2, and LMNB1, thus leading to cell cycle arrest and senescence-like state of the fibroblasts in the early phase.

**Characterization of fibroblasts treated with highly concentrated trehalose**. To further explore the effects of highly concentrated trehalose, we studied morphological alterations of fibroblasts after trehalose treatment. Phase contrast microscopy revealed dose-dependent morphological differences between fibroblasts cultured with or without trehalose. We observed that the shape of cells cultured with trehalose remained polygonal/expanded, although the control cells became fusiform/elongated (Fig. 5a and Videos 1–3). In Fig. 5a and Videos 1–3, we demonstrate that trehalose inhibited the population growth of monolayer fibroblast cells. Further, to clarify trehalose-induced cell proliferation inhibition, we examined cell viability via CCK8 assay. Trehalose slightly inhibited cellular proliferation of the fibroblasts (Supplementary Fig. 8). Importantly, instead of trehalose, we observed that high-concentration sucrose (100 mg/ml) in the medium inhibited cell proliferation and induced

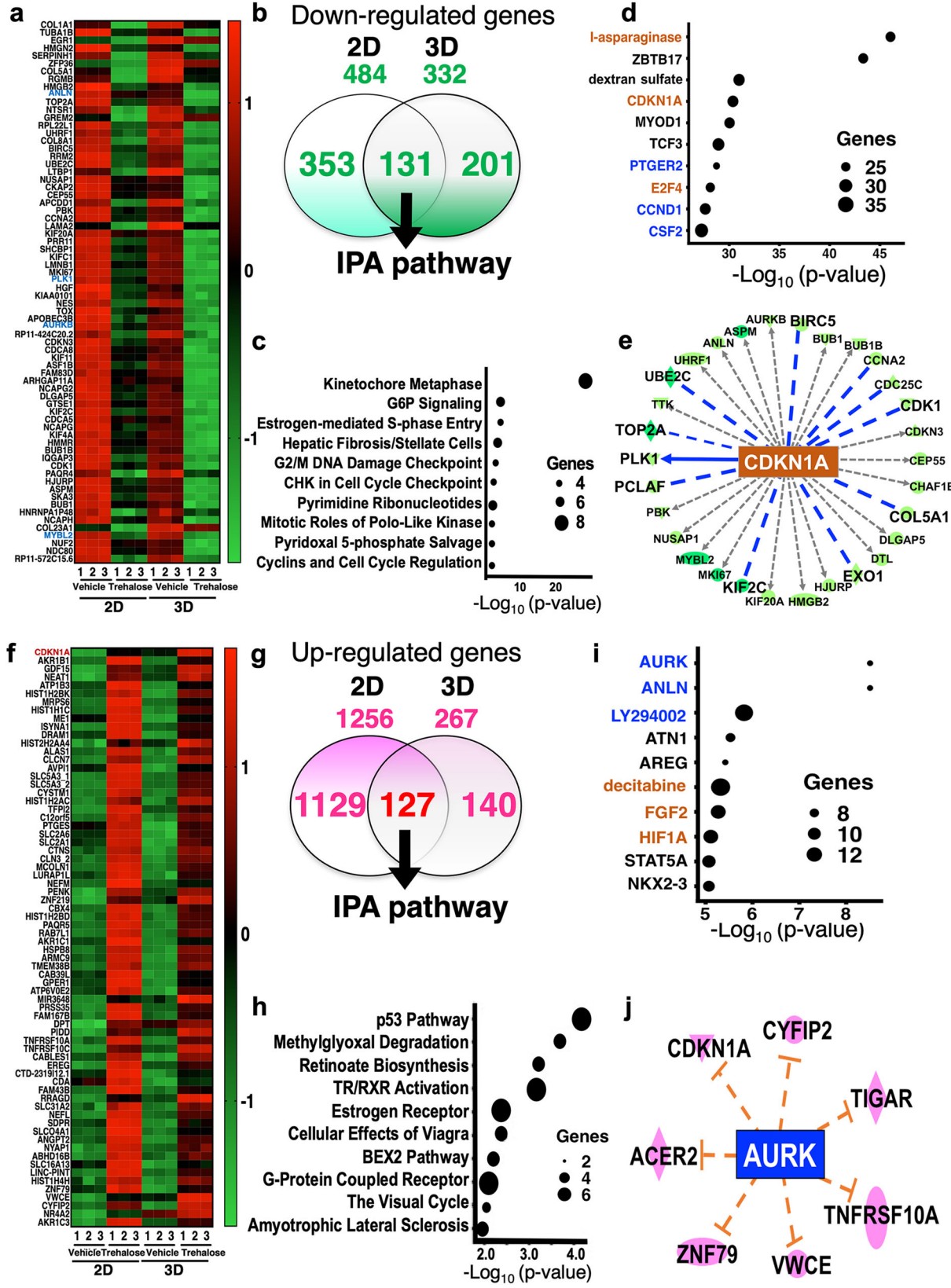

cell death in human dermal fibroblasts (Supplementary Fig. 8 and Video 4). Fibroblasts were characterized for a potential senescent phenotype via SA-βGAL staining, which revealed that remarkably more trehalose-treated fibroblasts (87%) than vehicle-treated fibroblasts (14%) were SA-βGAL positive (Fig. 5b). In addition, we observed that both high-concentration trehalose and vehicle

treatments induced similar DNA damage in fibroblasts, whereas hydrogen peroxide treatment induced remarkably more DNA damage than trehalose, evidenced by monitoring H2AX with phosphorylated Ser-139 (γH2AX) as a DNA damage marker (Fig. 5b). Similarly, flow cytometric analysis revealed that the percentage of cells showing DNA damage differed nonsignificantly

**Fig. 2 Highly concentrated trehalose-induced signaling pathways detected in human fibroblasts by whole transcriptome analysis with RNA-seq. a** The heatmap shows the *z*-scores of the gene expression downregulated in 2D and 3D fibroblasts by trehalose treatment for 24 and 72 h, respectively. **b** The Venn diagram demonstrates the numbers of downregulated genes. The 131 genes downregulated in 2D and 3D fibroblasts were used for an Ingenuity pathway analysis (IPA). **c** Canonical pathways detected in IPA using the genes downregulated by trehalose were shown together with the number of genes involved in the detected pathway. **d** Upstream factors of the downregulated genes by trehalose were indicated together with the number of genes involved in the detected factor. The upstream factors signaling by activation or by inhibition were shown in brown or blue, respectively. **e** A network demonstrates the interaction of CDKN1A, which was detected as an upstream factor of the downregulated genes, and the signals inhibited by trehalose, which are shown as blue lines. The green shapes indicate the genes downregulated by trehalose. **f** A heatmap shows the z-scores of the gene expressions upregulated in 2D and 3D fibroblasts by trehalose. **g** The Venn diagram demonstrates the numbers of the upregulated genes. **h** Canonical pathways detected in IPA using the genes upregulated by trehalose were shown together with the number of genes involved in the detected pathway. **i** Upstream factors of the upregulated genes by trehalose were indicated together with the number of genes involved in the detected factors. The upstream factors signaling by activation or by inhibition were shown in brown or blue, respectively. **j** A network demonstrates the interaction of AURK, detected as an upstream factor of the downregulated genes, and the factors predicted to activate interaction with AURK by trehalose, which are shown as red lines. The red shapes indicate the genes upregulated by trehalose.

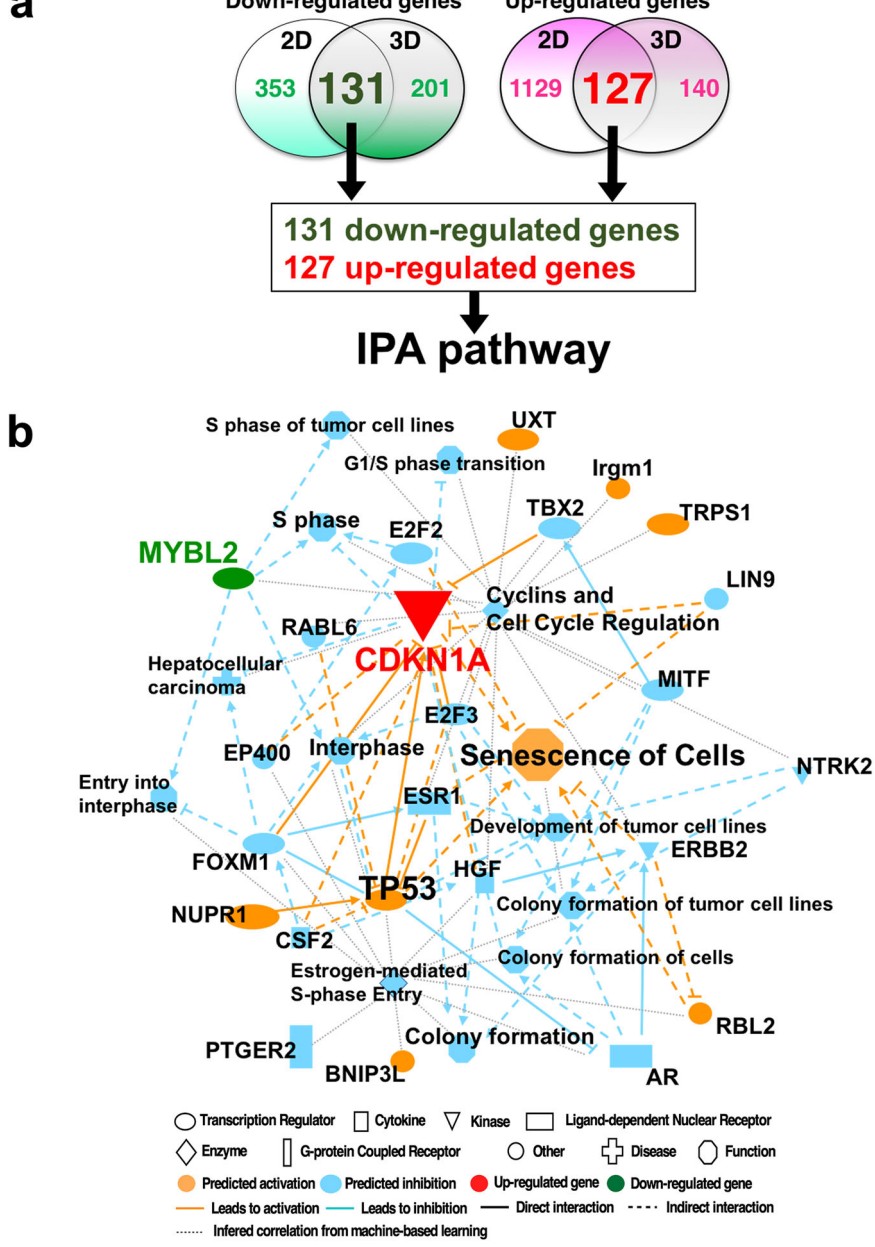

**Fig. 3 Graphical summary and network analysis by IPA pathway using genes modulated by trehalose. a** Venn diagram demonstrates the numbers of the downregulated genes and the upregulated genes analyzed in Fig. 2. There were 131 downregulated genes and 127 upregulated genes for an IPA. **b** A graphical summary shows the senescence cells, which are induced by p53 and CDKN1A, connected by the network analysis to the cellular functions.

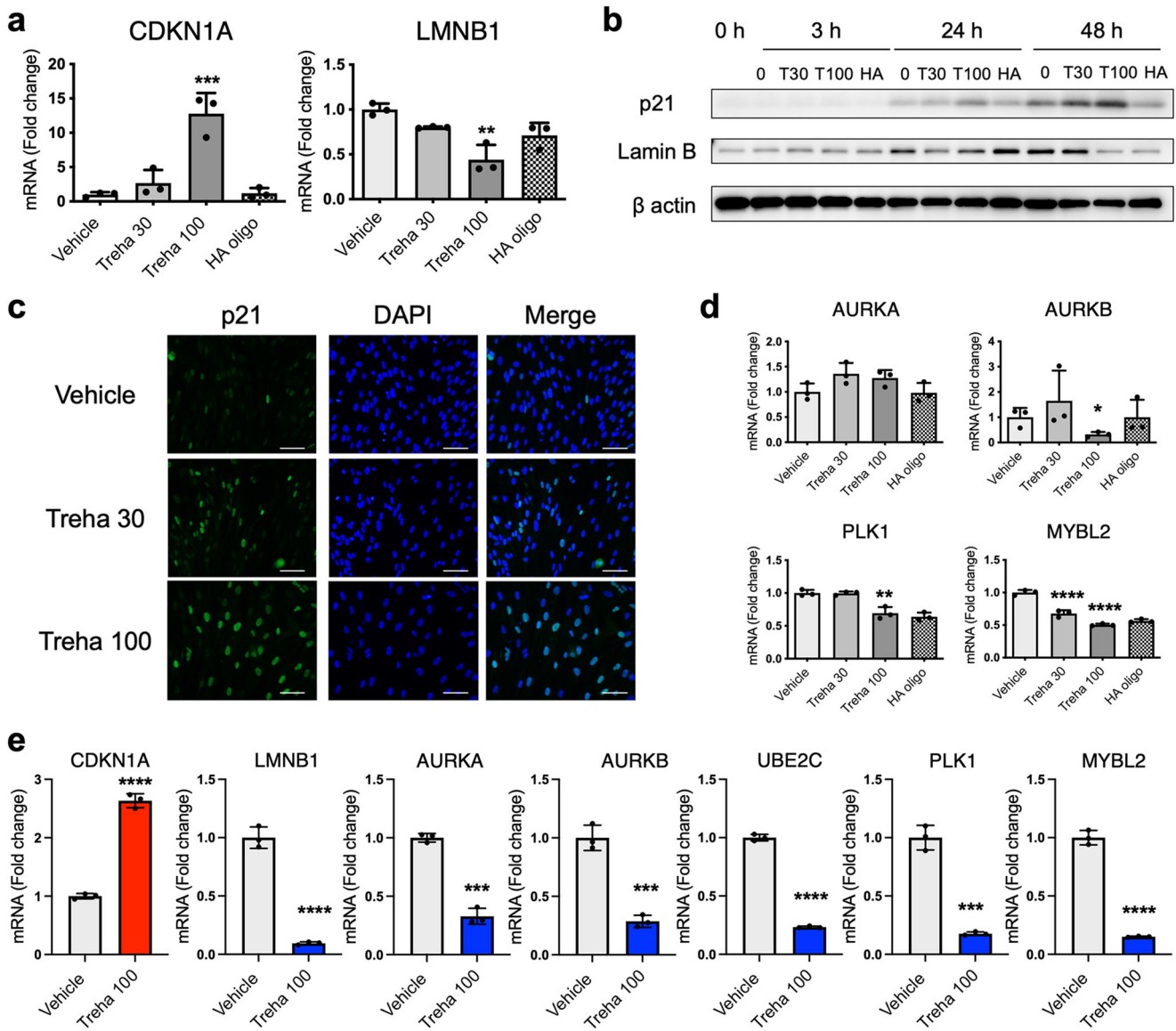

**Fig. 4 Trehalose modulates the expression of senescence and cell cycle arrest-related molecules. a** Human dermal fibroblasts were treated with trehalose (30 and 100 mg/ml), tetrasaccharide hyaluronan (HA oligo) (30 μg/ml), or vehicle (PBS) for 24 h. *CDKN1A* and *LMNB1* mRNA expression were assessed by qPCR. Data were shown as relative expressions to the control (vehicle-treated) fibroblasts. **b** Western blotting showing the expression of p21, lamin B, and β-actin in human dermal fibroblasts treated with trehalose (30 and 100 mg/ml) or the vehicle for 24 h. **c** Human dermal fibroblasts were treated with trehalose (30 and 100 mg/ml) or vehicle control (PBS) for 24 h. The cells were stained with antibodies for p21 (green) and DAPI (blue) for nuclei and were observed using a fluorescence microscope. Scale bar = 100 μm. **d** *AURKA*, *AURKB*, *AURKC*, *MYBL2*, *PLK1*, and *UBE2C* mRNA expressions were assessed by qPCR. Data were shown as the relative expression to the control (vehicle-treated) fibroblasts. **e** Trehalose (100 mg/ml) or vehicle (PBS) were added in the human dermal fibroblasts populated collagen gel for 72 h. *CDKN1A*, *LMNB1*, *AURKA*, *AURKB*, *UBE2C*, *PLK1*, and *MYBL2* mRNA expressions were assessed by qPCR. Data were shown as relative expression to the control (vehicle-treated). *$P < 0.05$, **$P < 0.01$, ***$P < 0.001$, and ****$P < 0.0001$ versus the vehicle-treated control group by one-way ANOVA (**a**, **d**) or Student *t*-test (**e**). Data were expressed as means ± SD for three wells (**a**, **d**) or three dermal substitutes (**e**), and are representative of three independent experiments.

between high-concentration trehalose and vehicle treatments, but they increased significantly after hydrogen peroxide treatment when γH2AX was monitored (Fig. 5c, d). These findings indicate that high-concentration trehalose induces the cytostatic effect and senescence-like state with dramatically less DNA damage than stressors that do promote senescence, such as hydrogen peroxide.

Next, fibroblasts treated with or without trehalose for 24 h were further investigated by Western blotting. High-concentration trehalose activated ERK1/2 and AKT (Fig. 5e). To further analyze the effect of trehalose on cell cycle progression, fibroblasts treated with or without trehalose for 24 h were analyzed by flow

cytometry after propidium iodide staining. Of the cells treated with trehalose (100 mg/ml) for 24 h, 30% were detected in the G2/M-phase, whereas 20% of the cells treated without trehalose were detected in the G2/M-phase (Fig. 5f). To further characterize the trehalose-treated cells detected in the G2/M-phase, we stained them with BrdU and analyzed them by flow cytometry using a dye that binds to total DNA (7-AAD). Significantly more cells were detected in the G2/M-phase 24 h after treatment with highly concentrated trehalose (Fig. 5g, h), which also significantly decreased the number of cells in the S phase (Fig. 5g, h). In addition, human dermal fibroblasts in the LSEs were stained with

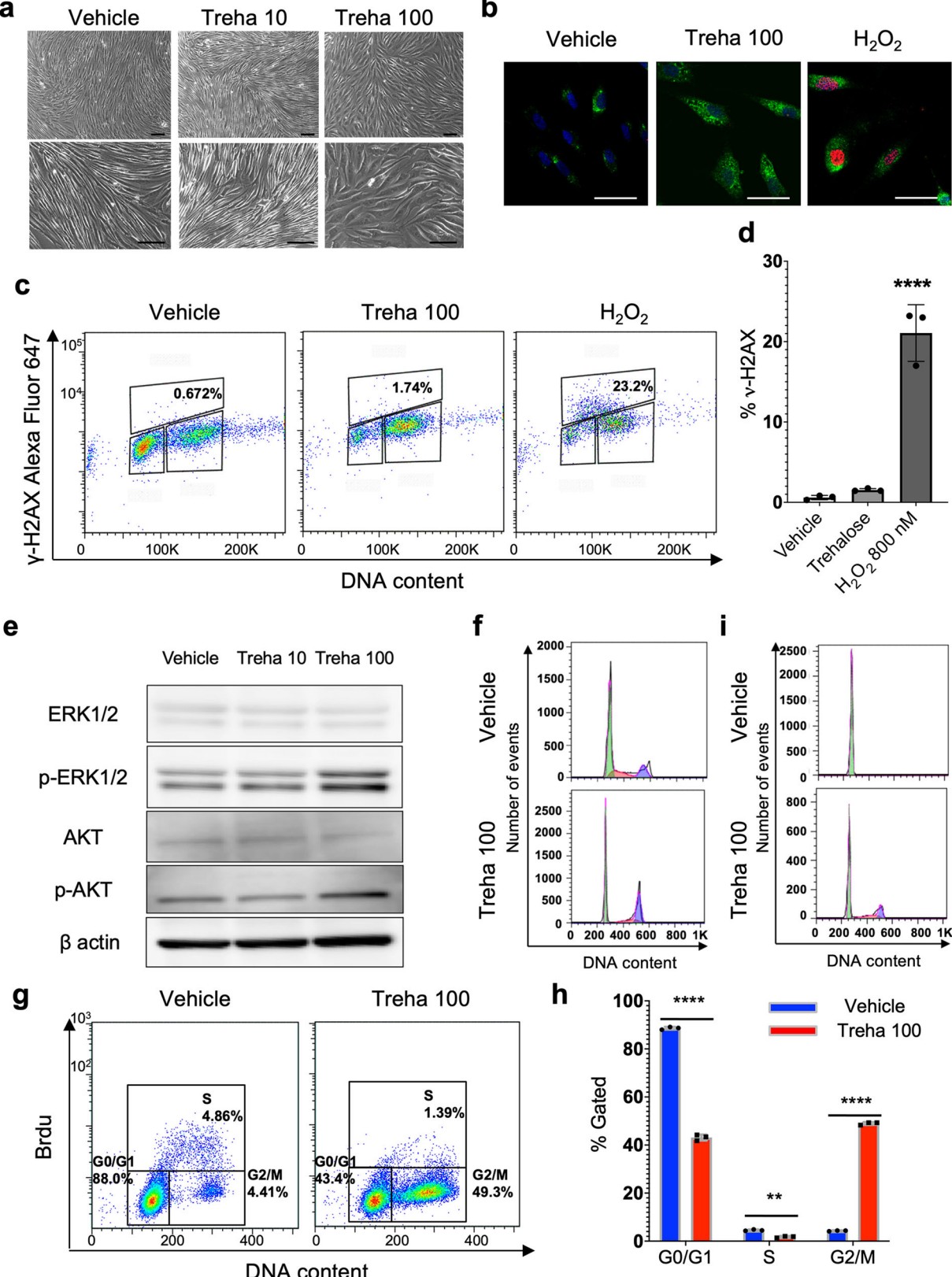

propidium iodide 7 days after air exposure, and the percentage of G2/M cells was measured by flow cytometry. Interestingly, 32% of cells cultivated with trehalose (100 mg/ml) were detected in the S and G2/M-phase, whereas almost none of the cells cultivated without trehalose accumulated in the S and G2/M-phase (Fig. 5i). These data indicate that trehalose sustained the proliferation of fibroblasts in 3D culture whereas vehicle treatment arrested them in the G1 phase after 1 week. Furthermore, significantly higher amounts of superoxide radicals were generated in trehalose-treated fibroblasts (Supplementary Fig. 9). Therefore, we conclude that trehalose triggers two antagonistic cell cycle regulatory pathways in fibroblasts: the classical mitogenic ERK and AKT

**Fig. 5 Trehalose arrests fibroblasts in the G2/M-phase of the cell cycle and triggers a senescence-like state with activation of Erk1/2 and Akt.**
**a** Representative photos of human dermal fibroblasts treated with trehalose (10 and 100 mg/ml) or vehicle (PBS) for 24 h. Phase contrast micrographs, bar = 50 μm. **b** Fibroblasts were characterized for a potential senescent phenotype via SPiDER-βGAL (green), γH2AX (red), and DAPI (blue) 24 h after treatment with trehalose (100 mg/ml), vehicle, or hydrogen peroxide (800 nM). Scale bar = 50 μm. **c, d** Flow cytometric analysis of DNA damage was performed with the DNA Damage Detection Kit to monitor H2AX with phosphorylated Ser-139 (γH2AX) as a DNA damage marker. Alexa Fluor 647-conjugated anti-γH2AX and propidium iodide (PI) staining 24 h after vehicle, hydrogen peroxide (800 nM), or trehalose (100 mg/ml) treatment (**c**), and the numerical analysis was conducted for each experiment (**d**). Data were expressed as means ± SD and are representative of three independent experiments. **e** Western blotting showing the expression of ERK1/2, p-ERK1/2, AKT, p-AKT, and β-actin in human dermal fibroblasts treated with trehalose (10 and 100 mg/ml) or vehicle for 24 h. **f** Twenty-four hours after the trehalose (100 mg/ml) or vehicle treatment, human dermal fibroblasts were stained with PI, and the percentage of G2/M cells was measured by flow cytometry. **g, h** Flow cytometric analysis of cell cycle distribution was performed with FITC-conjugated anti-BrdU and 7-AAD staining 24 h after vehicle or trehalose (100 mg/ml) treatment (**g**), and numerical analysis was conducted for each experiment (**h**). Data were expressed as means ± SD and are representative of three independent experiments. **i** Seven days after air exposure, human dermal fibroblasts in the living skin equivalent were stained with PI, and the percentage of G2/M cells was measured by flow cytometry. Representative FACS images. Data were representative of two independent experiments.

pathway and a novel G2/M cell cycle arrest pathway with induction of p21 without the induction of DNA damage.

**Upregulation of wound healing-related genes in trehalose-treated fibroblasts.** Senescent cells exhibit a hypersecretory phenotype, which has been referred to as the SASP[11]. The SASP comprises a collection of growth factors[11,21]. The beneficial and transient initiation of senescence could contribute significantly to a cutaneous wound healing process[13]. Next, we aimed to elucidate the characteristics of these high-concentration trehalose-induced senescence-like state in human fibroblasts. RNA-seq analysis data and the IPA of the differentially expressed genes revealed significant upregulation of ten wound healing-related genes—*EREG, ARG2, CCL2, IL1RN, PGF, SPP1, VEGFA, FGF2, ANGPT2*, and *DPT*. Then, to confirm RNA-seq findings, qPCR mRNA expression analysis of the wound healing-related genes was performed, which revealed a significant increase in fibroblasts treated with trehalose (30 or 100 mg/ml) for 24 h compared with vehicle- or HA-treated control fibroblast (Fig. 6a). We confirmed these effects of trehalose using the fibroblasts derived from three other patients. DPT has a vital role in promoting keratinocyte migration during re-epithelialization in wound healing[22]. DPT secretion in the medium and DPT expression in the fibroblasts were assessed by ELISA and Western blotting, respectively. We found a significant increase in DPT protein secretion in the cultured medium 48 h after trehalose treatment compared with that of the vehicle-treated control fibroblasts by ELISA (Fig. 6b). Interestingly, results from Western blotting demonstrated trehalose treatment increased the remarkable DPT protein expression in cell lysate after 48 h (Fig. 6c). Consistent with these findings, qPCR mRNA expression analysis of *DPT* in the 3D fibroblasts, which were embedded in the collagen gel with trehalose (100 mg/ml) for 72 h, revealed a significant increase compared with that of the vehicle-treated fibroblasts (Fig. 6d). Furthermore, RNA-seq analysis of these 3D fibroblasts revealed 267 upregulated genes in the trehalose-treated fibroblasts compared with those of the vehicle-treated control fibroblasts, including significant upregulation of nine wound healing-related genes—*EREG, ARG2, CCL2, IL1RN, PGF, SPP1, VEGFA, FGF2*, and *ANGPT2*. Then, to confirm RNA-seq findings, qPCR mRNA expression analysis of these wound healing-related genes was performed; this revealed a significant increase in the 3D fibroblasts that were embedded in the collagen gel with trehalose (100 mg/ml) for 72 h compared with that of the control fibroblasts (Fig. 6d).

**CDKN1A is involved in the upregulation of SASP factor genes.** Next, we investigated the involvement of CDKN1A in trehalose-induced upregulation of SASP-related genes in 2D fibroblasts.

CDKN1A siRNA transfection significantly suppressed its mRNA levels, confirming its knockdown (Fig. 7). CDKN1A knockdown significantly suppressed the upregulation of *DPT, ANGPT2, VEGFA, EREG*, and *FGF2* after trehalose treatment (Fig. 7), demonstrating the critical role of CDKN1A in effecting the trehalose-induced senescence-like state in fibroblasts.

**Effect of high-concentration trehalose on the expression of noninflammatory SASP-related factor genes.** The SASP is closely associated with positive and negative outcomes depending on cell types and contexts. Senescent cells also generate proinflammatory molecules and matrix metalloproteinases. Next, we investigated whether the trehalose-induced cellular senescence-like state in monolayer and organotypic cultures of human fibroblasts would lead to similarly dramatic changes in senescence-related factors as induced by other stressors. Surprisingly, analysis of RNA-seq data revealed that SASP factor genes related to inflammation, such as *IL-6, IL-8*, and *IL-1B*, were not elevated, whereas several genes associated with senescence, such as *GDF15, MMP3*, and *TNFRSF10C*, were upregulated (Supplementary Table 1). Overall, our data support the hypothesis that highly concentrated trehalose elicits a noninflammatory senescence-like state.

**Dermal substitute with high-concentration trehalose-treated fibroblasts enhances wound closure and capillary formation.** Since trehalose treatment upregulated wound healing-related genes in the 3D culture system, 6 mm × 6 mm full-skin thickness excisional wounds were made on the dorsum of nude mice (BALB/cAJcl-nu). The dermal substitutes composed of collagen and fibroblasts with or without trehalose (100 mg/ml) were further transplanted to test whether trehalose-treated fibroblasts in the dermal substitute accelerate wound closure in vivo. In the macromorphological analysis, we detected a significant tendency toward promoted healing in the high-concentration trehalose-treated group compared with the control group (Fig. 8a, b). Induction of neoangiogenesis was observed 7 days after transplantation of the dermal substitute with high-concentration trehalose (Fig. 8c). Furthermore, statistical analysis demonstrated a significant difference in the narrower wound opening of the trehalose-treated dermal substitute group as detected histologically by hematoxylin and eosin staining 7 days after transplantation (Fig. 8d, e). We also stained the tissue with antibodies against CD31. Compared with that in the vehicle-treated group, a significantly greater number of vessels in the wounds transplanted with the trehalose-treated dermal substitute stained positive for CD31 (Fig. 8f, g). Therefore, the dermal substitute with trehalose-treated fibroblasts accelerated wound healing by promoting capillary formation in vivo.

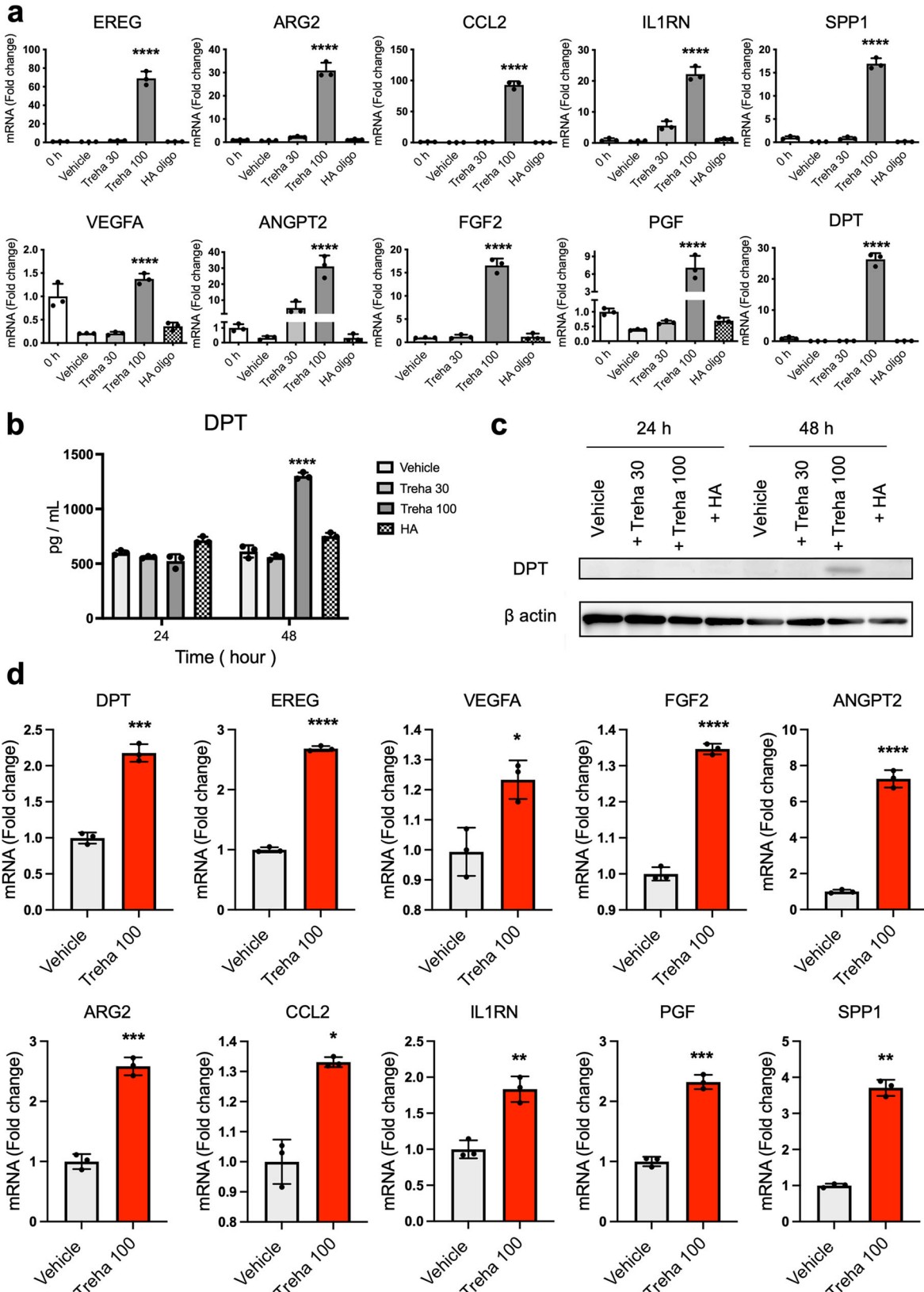

**CDKN1A is involved in the facilitated wound healing by the dermal substitute with high-concentration trehalose-treated fibroblasts.** Next, we investigated the involvement of CDKN1A in the trehalose-induced promotion of wound healing in vivo. CDKN1A knockdown was confirmed by its significantly suppressed mRNA levels post-siRNA transfection (Supplementary Fig. 10).

Full-skin thickness excisional wounds of 6 mm × 6 mm were made on the dorsum of nude mice (BALB/cAJcl-nu), and the dermal substitutes composed of collagen and fibroblasts (control or CDKN1A siRNA-treated) with or without trehalose (100 mg/ml) were transplanted to test whether trehalose treatment accelerates in vivo wound closure via CDKN1A. Macromorphological analysis

**Fig. 6 Trehalose induces an increase in the expression of wound healing-related molecules. a** Human dermal fibroblasts were treated with trehalose (30 and 100 mg/ml), tetrasaccharide hyaluronan (HA oligo) (30 μg/ml), or vehicle control (PBS) for 24 h. *EREG*, *ARG2*, *CCL2*, *IL-1RN*, *PGF*, *SPP1*, *VEGF*, *ANGPT2*, and *DPT* mRNA expressions were assessed by qPCR. Data were shown as relative expression to the control (0 h) fibroblasts. For *FGF2* mRNA expression, data are shown as relative expression to vehicle control groups at 24 h. **b** DPT was measured by ELISA in the culture medium of human dermal fibroblasts. One set of fibroblasts was treated with trehalose 30 or 100 mg/ml, tetrasaccharide hyaluronan (HA) (30 μg/ml), or the vehicle (PBS) for 24 or 48 h (n = 3). **c** Representative Western blots showing DPT and β-actin expression in human dermal fibroblasts 24 or 48 h after vehicle, trehalose 30 or 100 mg/ml, tetrasaccharide hyaluronan (HA) (30 μg/ml) exposure. **d** Trehalose (100 mg/ml) or vehicle were added in the human dermal fibroblasts populated collagen gel for 72 h. *DPT*, *EREG*, *VEGF*, *FGF2*, *ANGPT2*, *ARG2*, *CCL2*, *IL-1RN*, *PGF*, and *SPP1* mRNA expression were assessed by qPCR. Data were shown as the relative expression to the control (vehicle-treated). *$P < 0.05$, **$P < 0.01$, ***$P < 0.001$, ****$P < 0.0001$ versus the vehicle-treated control group by one-way ANOVA (**a**, **b**) or Student *t*-test (**d**). Data were expressed as means ± SD for three wells (**a**, **b**) or three dermal substitutes (**d**), and representative of three independent experiments.

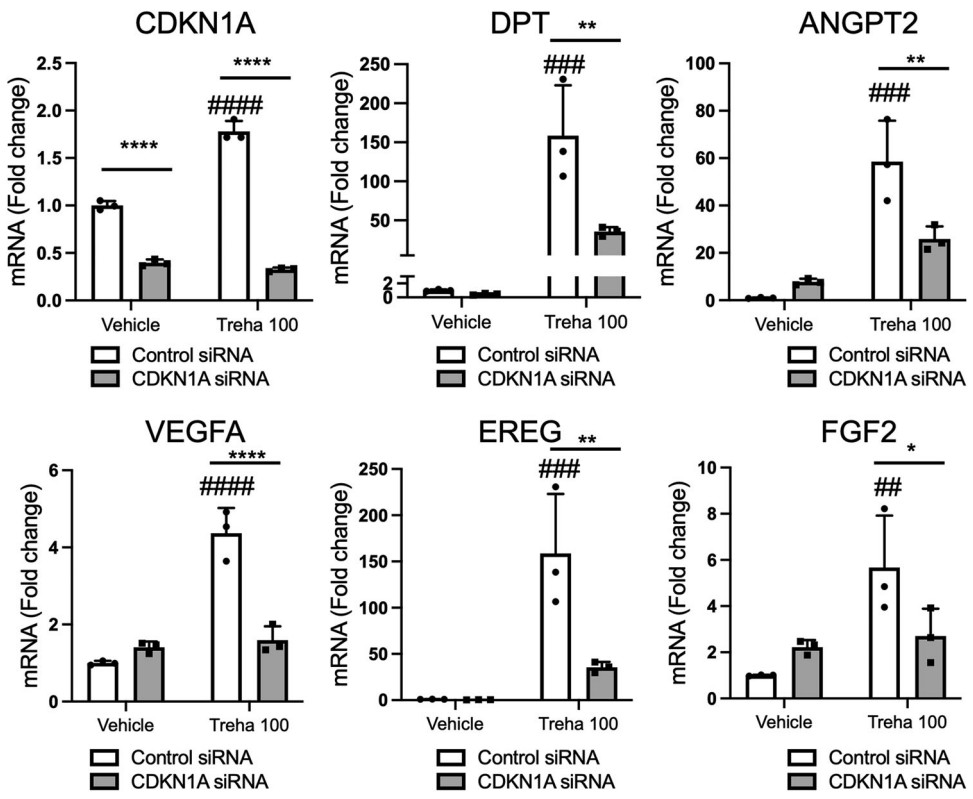

**Fig. 7 CDKN1A is involved in a trehalose-induced increase in the expression of wound healing-related molecules.** After transfection with control or *CDKN1A* siRNA, human dermal fibroblasts were treated with trehalose (100 mg/ml) or vehicle control (PBS) for 48 h. *CDKN1A*, *DPT*, *ANGPT2*, *VEGF*, *EREG*, and *FGF2* expressions were assessed by qPCR. Data were shown as relative expression to the control siRNA- and vehicle-treated fibroblasts. Data were expressed as means ± SD and are representative of two independent experiments with similar results. *$P < 0.05$, **$P < 0.01$, and ****$P < 0.0001$ versus the relevant control group, and ##$P < 0.01$, ###$P < 0.001$, and ####$P < 0.0001$ versus the control siRNA-treated group of vehicle-treated fibroblasts by two-way ANOVA.

revealed that CDKN1A knockdown significantly suppressed wound healing in the high-concentration trehalose-treated group compared with the control group (Fig. 9a, b). Neoangiogenesis was not induced 7 days posttransplantation in the high-concentration trehalose- and CDKN1A siRNA-treated group (Fig. 9c). Furthermore, hematoxylin and eosin staining demonstrated a nonsignificant difference in the narrower wound opening of the trehalose-containing dermal substitute group 7 days posttransplantation after treatment with CDKN1A siRNA (Fig. 9d, e). Compared with the vehicle-treated group, a nonsignificantly greater number of vessels in the wounds transplanted with the trehalose- and CDKN1A siRNA-treated fibroblasts stained positive for CD31 (Fig. 9f, g), suggesting the critical role of CDKN1A in trehalose-induced promotion of wound healing in vivo.

### Discussion

The challenge to engineer cultured epidermal autografts for the life-saving treatment of patients with extensive, full-thickness burns was accomplished using the method of keratinocyte-cultivation described by Rheinwald and Green[23,24]. However, their method requires the use of a feeder layer of lethally irradiated mouse 3T3 cells and serum. Therefore, regulatory issues have necessitated the use of xenotransplantation and the development of cultivation technology. Moreover, epidermal grafts without the dermis are less resistant to trauma and more prone to posttransplantation contracture, leading to poor functional and cosmetic outcomes. An autograft full-thickness LSE cannot be used to treat patients with burns due to the time required for preparation, despite the advances in the methods for rapid ex vivo expansion. In this report, we demonstrated a breakthrough in the new techniques for the rapid development of LSE using the effect of highly concentrated trehalose added to collagen gel, even as a pretreatment, which induces the transient beneficial senescence-like state in human fibroblasts via CDKN1A/p21 and modulates the capacity to accelerate the proliferation of keratinocytes in the epidermal layer of LSE. Furthermore, the trehalose-treated skin

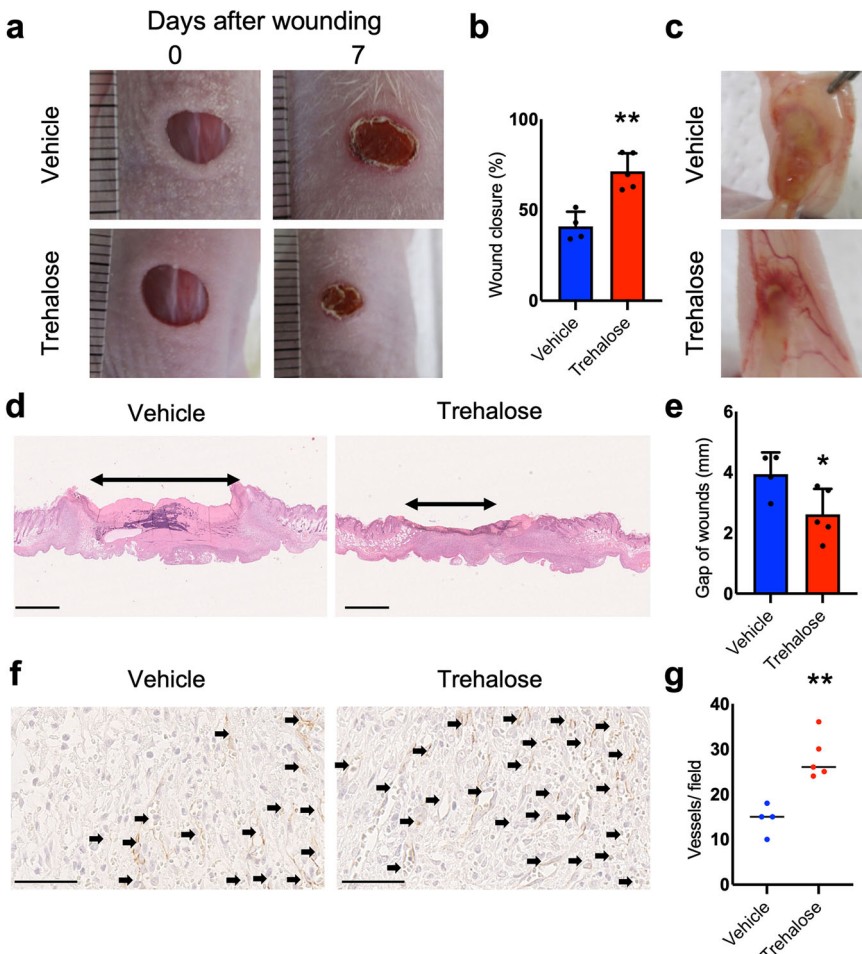

**Fig. 8 Graft of dermal substitute with highly concentrated trehalose-treated fibroblasts on nude mice accelerates murine wound closure and angiogenesis. a** Representative photographs of the wound area on the recipient nude mice as indicated at 7 days after transplantation. **b** Wound closure was quantified and presented as % wound closure; % = the percentage of the initial wound area size at day 7 when comparing the trehalose-LSE group (red band) to the control group (blue band). **c** Representative photographs of angiogenesis induced by the grafts. **d** Images of the H&E-stained tissue sections from the wound sites on day 7. Bars = 1 mm. **e** Quantitative analysis of the gap of the wounds in comparison between the trehalose-LSE group (red band) and control group (blue band). **f** Images of CD31 immunostaining on the day 7 wounds. Bars = 50 μm. **g** Quantitative analysis of CD31-positive vessels per field. Data were expressed as means ± SD. A subsequent statistical analysis was performed with the Student $t$-test. *$P < 0.05$ and **$P < 0.01$, with four mice in the control group, five in the trehalose-LSE group, and one tissue section from each mouse. Data were representative of three independent experiments.

equivalents promoted wound repair with an angiogenic effect in vivo via CDKN1A/p21 (Fig. 10).

These observations elucidate important physiologic roles for fibroblasts in the construction of LSE in vitro and in wound repair in vivo. Our 3D culture system confirmed the role of trehalose-treated fibroblasts in keratinocyte proliferation. Somewhat surprisingly, we observed the significant upregulation of growth factors such as DPT, FGF2, EREG, VEGF, and ANGPT2 as growth factors in the highly concentrated trehalose-induced senescence-like state. Importantly, we discovered that this effect of trehalose was transient and occurred only at an early time point, because RNA-seq analysis revealed similar gene expression profiles for trehalose-treated and vehicle-treated keratinocytes and fibroblasts in the final LSE preparations. Furthermore, we found that a trehalose-induced senescence-like state was relatively noninflammatory compared with other stress-induced SASPs. Therefore, the induction of a trehalose-induced transient noninflammatory senescence-like state of human fibroblasts is a novel approach for accelerating keratinocyte proliferation in LSE and wound healing in vivo. These grafts containing the fibroblasts cultivated with highly concentrated trehalose may have a major

impact on chronic wound therapy. We observed that the addition of sucrose of the same concentration (100 mg/ml) in the medium induced cell death in human dermal fibroblasts, suggesting that these effects of the trehalose is not due to the stress of the disaccharide-induced osmotic pressure.

The use of RNA-seq technology enables an unbiased, sensitive method for investigating the transcriptome of LSEs, 3D fibroblasts, and the 2D monolayer under trehalose treatment. We identified significantly and differentially expressed genes overlapping between the two data sets for 2D and 3D fibroblasts. RNA-seq analysis data and the IPA of the differentially expressed genes revealed a potential key role for the CDKN1A pathway for transient G2 arrest based on the discovery of the upregulation of other growth factors, including DPT. The *CDKN1A* gene represents a major target of p53 activity, and its product, p21, is the major regulator of the cellular stress response[25]. The capacity of p21 for cell cycle arrest depends upon its nuclear localization[26]. Our immunocytochemistry results revealed that p21 levels increased mainly in the nucleus after trehalose treatment. Additionally, the RNA-seq data revealed inhibition of AURKA and AURKB in the fibroblasts treated with trehalose. The Aurora kinases, including AURKA and AURKB, are highly

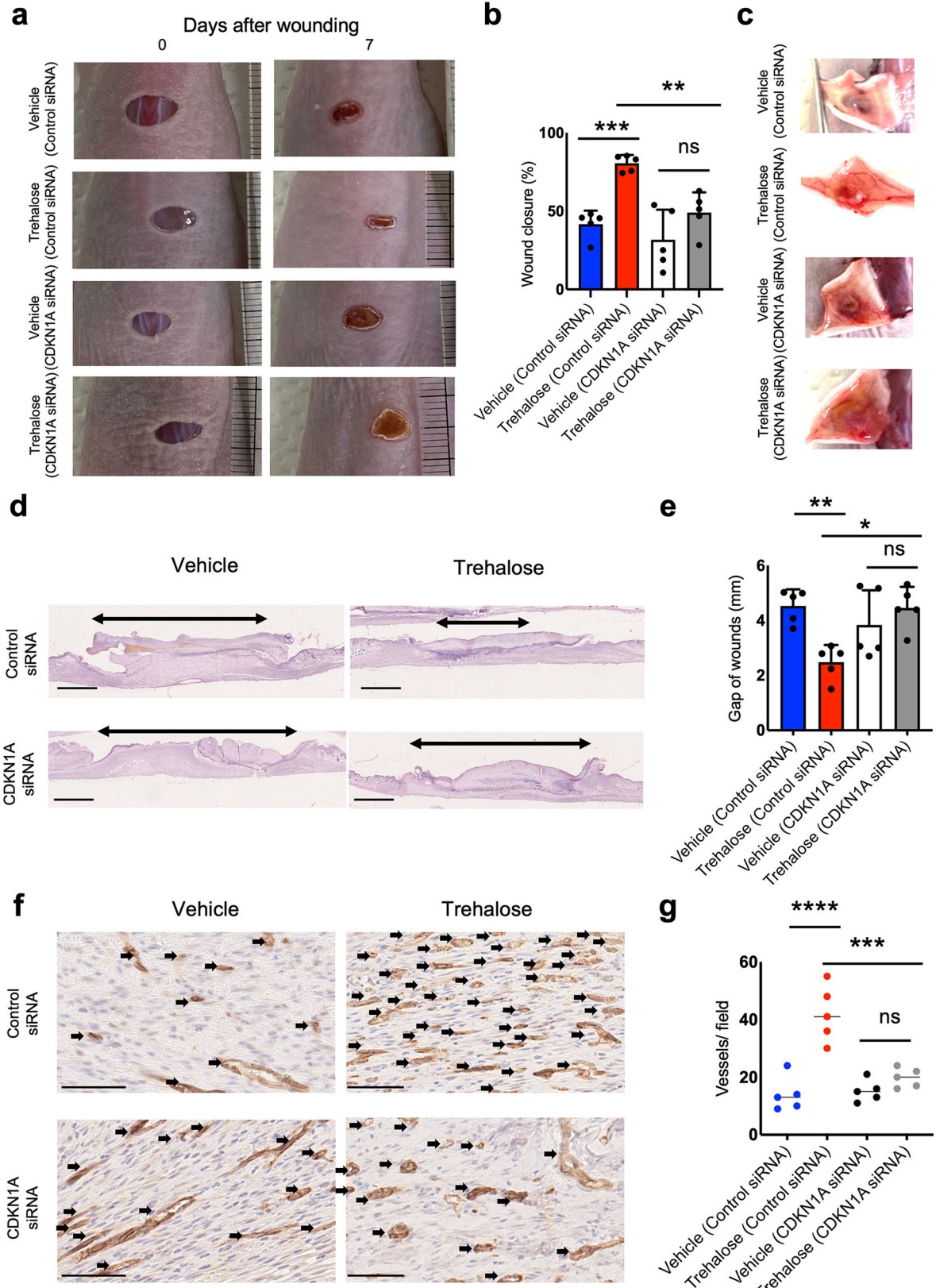

conserved serine/threonine kinases essential for the control of mitosis. AURKB indirectly represses the expression of CDKN1A at the transcriptional level[27]. Thus, the downregulation of AURKB leads to the induction of p21. Treatment of cultured multiple myeloma cells with MLN8237, which is a small-molecule, and Aurora-A kinase inhibitor that inhibits Aurora-A gene expression

by siRNA, results in G2/M arrest and senescence in vitro[28]. P21 inhibits AURKA by regulating E2F3[29]. Hence, under the stress of high-concentration trehalose, induced transcription of *CDKN1A* inhibited AURKA, which led to the G2/M blockade (CDK1-induced mitotic entry). Furthermore, upstream analysis of RNA-seq data revealed the involvement of ZBTB17, which binds to the

**Fig. 9 Graft of dermal substitute with highly concentrated trehalose-treated fibroblasts on nude mice accelerates wound closure and angiogenesis via CDKN1A. a** Representative photographs of the wound area on the recipient nude mice as indicated at 7 days after transplantation. **b** Wound closure was quantified and presented as % wound closure; % = the percentage of the initial wound area at day 7 when comparing the trehalose-LSE group (red band) to the control group (blue band). **c** Representative photographs of angiogenesis induced by the grafts. **d** Images of the H&E-stained tissue sections from the wound sites at day 7. Bars = 1 mm. **e** Quantitative analysis of the gap of the wounds in comparison between the trehalose-LSE group (red band) and control group (blue band). **f** Images of CD31 immunostaining on day 7 wounds. Bars = 50 μm. **g** Quantitative analysis of CD31-positive vessels per field. Data were expressed as means ± SD. A subsequent statistical analysis was performed with one-way ANOVA. *$P < 0.05$, **$P < 0.01$, with five mice in each group, and one tissue section from each mouse. Data were representative of two independent experiments.

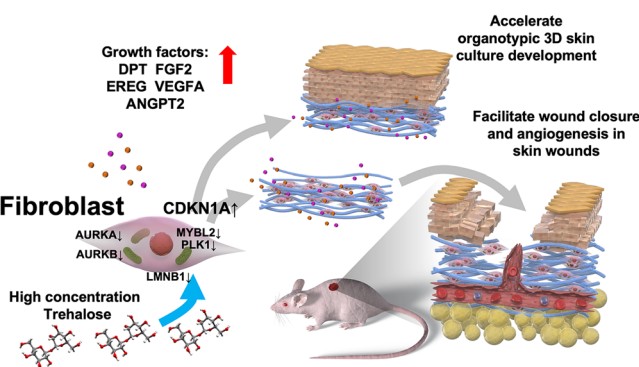

**Fig. 10 Trehalose-induced senescence-like state.** Highly concentrated trehalose-induced transient senescence-like state accelerates organotypic skin culture generation and facilitates cutaneous wound healing.

*AURKA* promoter and is assumed to be associated with transcriptional factors that induce *AURKA* downregulation following topoisomerase I inhibition[30].

DeBosch et al. reported that the activity of glucose transporters at the plasma membrane (SLC2A1: GLUT1, SLC2A2: GLUT2, SLC2A3: GLUT3, SLC2A4: GLUT4, and SLC2A8: GLUT8), which are expressed in fibroblasts[31], is inhibited by 100-mM trehalose[32]. We treated the fibroblasts with 29.2- (10 mg/ml) to 292-mM (100 mg/ml) trehalose and observed a dose-dependent effect in our system. Upon glucose restriction in the medium (4.4 mM), yeast cells underwent transient cell cycle arrest at the G2 phase, which is dependent on the Wee1 tyrosine kinase[33]. We identified a significant decrease in *WEE1* mRNA levels in trehalose-treated 2D and 3D fibroblasts compared with that of the control in our RNA-seq data. Therefore, trehalose-mediated inhibition of glucose absorption in human dermal fibroblasts might lead to transient Wee1-independent G2 cell cycle arrest in our system. We also identified increased mRNA levels of *HIF1A* after trehalose treatment in the cultured monolayer and 3D fibroblasts. Interestingly, the levels of reactive oxygen species (ROS) in both the mitochondria and the cytosol increased upon glucose withdrawal, although reduced to the background level upon glucose refeeding in human newborn foreskin fibroblasts[34]. There was significantly higher superoxide radical generation in those fibroblasts treated with trehalose (Supplementary Fig. 9). The growing body of evidence suggests that ROS induces hypoxia-inducible factor 1-α (HIF1α) via MAPK, ERK, and PI3K/AKT pathways[35]. HIF1α negatively regulates AURKA in breast cancer cell lines under hypoxic conditions[36] and is involved in *CDKN1A* transcription in murine embryonic fibroblasts[37]. Hence, elevated ROS levels after glucose transport inhibition in trehalose-treated fibroblasts induced HIF1α and could be involved in the subsequent transcription of *CDKN1A*, thus leading to a G2/M blockade in our system.

The role of CDKN1A is likely confined to the induction of senescence and cells can resume cycling upon resolution of stress[38]. In embryonic development, CDKN1A induction leads to

SASP factor expressions, like FGFs, which stimulate cell proliferation and tissue formation[39]. Importantly, we found highly concentrated trehalose can markedly increase CDKN1A/ p21 expression, which is required for a striking upregulation of FGF2 and other growth factors. Demaria et al. analyzed p16INK4a/ CDKN1A double knockout mice, and found that the wound healing of knockout mice was impaired compared with their wild-type controls, thus indicating that the presence of senescent cells facilitates skin wound healing, and their absence significantly suppresses wound closure[12]. Thus, transient senescence is critical for effective cutaneous wound healing. We hypothesize that trehalose treatment can induce a transient and beneficial senescent-like phenotype of the human fibroblasts via CDKN1A/p21 for optimal wound healing.

Previous studies demonstrated that epidermal growth factor (EGF) family members—transforming growth factor (TGF)-α, heparin-binding (HB)-EGF, and EREG—act as autocrine growth factors for normal human keratinocytes[40]. We also observed dramatic upregulation of *EREG* after trehalose treatment. EREG is upregulated in the psoriatic epidermis and was initially purified from the mouse NIH-3T3[40]. We also demonstrated that trehalose treatment induced a marked and significant increase of mRNA and protein levels of DPT, which is a 22-kDa matrix protein for the interaction with TGFβ1, fibronectin, and decorin. DPT dose-dependently promotes keratinocyte migration in wound repair[22]. Furthermore, trehalose attenuates protein aggregation and maintains polypeptide chains in a partially folded state for the refolding by cellular chaperones[41]. Therefore, we speculate that trehalose plays a role in inducing the secretion of growth factors from fibroblasts and facilitates the functions of these overexpressed growth factors as chaperones for keratinocyte proliferation on collagen gels.

Although CDKN1A is a vital senescence marker, it is induced during transient cell cycle arrest, and thus must be used in combination with other markers[16]. A recently emerging candidate marker that seems to play a role in attenuating senescence is MYBL2, which is a transcription factor of the MYB family[42]. The p53–p21 pathway suppresses MYBL2 expression as a stress response[43]. Our RNA-seq results demonstrated that downregulation of *MYBL2* and target genes transactivated by MYBL2, such as *AURKA*, *CCNA2*, *CCNB1*, *CDK1*, *PLK1*, and *TOP2A*, are present in human fibroblasts following trehalose treatment, which indicates that MYBL2 may participate in the senescence-like phenotype of human fibroblasts after trehalose treatment. Furthermore, reduced *LMNB1* mRNA expression strongly predicts the senescence phenotype. LMNB1 expression decreased in human fibroblasts after trehalose treatment. Thus, LMNB1 can be a marker for a trehalose-induced senescence-like state.

Pertinent roles for transient senescence in tissue injury have been identified during wound repair: PDGFA-enriched SASP[12]. SASP factors include CCL2, which is a chemokine required for the chemotaxis of macrophages and monocytes during angiogenesis in wound repair. Recently, Whelan et al. demonstrated a novel role of mesenchymal stromal cell-derived CCL2 in accelerated wound closure[44]. In addition, the absence of the IL-1 receptor antagonist

(IL1RN) impaired wound healing along with aberrant NF-κB activation and reciprocal suppression of the TGF-β pathway[45]. Placental growth factor (PGF) encodes a growth factor that is homologous to the VEGF and is a potent angiogenic/permeability factor during wound repair[46]. Osteopontin (OPN) is a glycoprotein that is encoded by secreted phosphoprotein 1 gene (SPP1), and analysis of OPN null mutant mice indicated the in vivo role of OPN in structural remodeling and resolution of dermal wounds[47]. Angiopoietin-2 encoded by the ANGPT2 gene acts as a Tie-2 antagonist[48], thus increasing sensitivity to other proangiogenic factors such as VEGF[49]. We demonstrated significant upregulation of these wound healing-related genes, such as EREG, CCL2, IL1RN, PGF, SPP1, VEGFA, DPT, FGF2, ARG2, and ANGPT2, using RNA-seq and qPCR mRNA expression analysis in trehalose-treated 2D and 3D fibroblasts. Furthermore, we found that transplantation of a dermal substitute prepared with collagen gel and fibroblasts treated with trehalose promoted wound healing and capillary formation in vivo compared with wound recovery of the vehicle-treated control. Considered together, trehalose-induced angiopoietin-2 may act in concert with these SASP factors, such as VEGFA, to stimulate angiogenesis in the wounds. Although the results are limited to the murine model, we provide evidence that induction of the transient noninflammatory senescence-like phenotype by trehalose in fibroblasts is beneficial for healing skin wounds. In a p53-induced senescence model, cooperation between p21 and Akt was required to induce the cellular senescence phenotype and cell cycle arrest[18]. Interestingly, we demonstrated that high-concentration trehalose activates ERK1/2 and AKT in human fibroblasts. By contrast, Wu et al. reported that intraperitoneally injected trehalose promotes the survival of rat skin flaps and angiogenesis by enhancing autophagy via Akt inhibition[50]. Thus, the mechanism of action of high-concentration trehalose is different from what was previously known. Moreover, PGF and VEGFA accelerate diabetic wound healing, so these gene transfers to diabetic wounds have received increasing attention[51,52]. Topical application of CCL2 can accelerate cutaneous wound healing in mice with diabetes by promoting neovascularization[53]. Therefore, future therapies with high-concentration trehalose might be promising for the treatment of chronic wounds of human diabetic patients.

In conclusion, this study demonstrated that highly concentrated trehalose induces a transient trehalose-induced senescence-like state in fibroblasts, and revealed that trehalose-induced cell cycle arrest and growth factor secretion via CDKN1A/p21 are beneficial for keratinocyte proliferation in LSE construction in vitro and capillary formation and wound closure in the repair process in vivo. These data suggest a new therapeutic approach for altering wound responses by applying trehalose-treated fibroblasts for accelerating wound repair. This could provide the foundation of a new therapy to treat chronic diabetic and venous ulcers. Therefore, we believe these findings should promote future studies on the effect of high-concentration trehalose in modulating fibroblast functions for LSE construction and subsequent wound therapy.

## Methods

**Chemicals and reagents**. Trehalose (containing >98.0% trehalose dihydrate, Hayashibara, Okayama, Japan), oligo-hyaluronan (hyaluronan oligosaccharide 4mer, CSR-11006, Cosmo Bio, Tokyo, Japan), hydrogen peroxide (FUJIFILM Wako Pure Chemicals Corporation, Osaka, Japan), and biotin-labeled hyaluronic acid-binding protein (HKD, Sapporo, Japan) were purchased. Further requests for reagents should be directed to the Lead Contact, Jun Muto (junmuto@m.ehime-u.ac.jp).

**Cell culture**. Normal human epidermal keratinocytes were isolated from normal human skin and cultured under serum-free conditions, following a previously described method[54,55]. The cells were used for LSE cultures in their fourth passage. Fibroblasts were isolated from normal human skin and cultured in Dulbecco's modified Eagle medium (DMEM) (Thermo Fisher Scientific) supplemented with 10% fetal calf serum, and fifth-passage cells were used to construct the LSEs. We isolated the cells from the skins of the extremities of two one-year-old boys and two

1-year-old girls. All procedures that involved human subjects received prior approval from the Ethics Committee of Ehime University School of Medicine, Toon, Ehime, Japan, and all subjects provided written informed consent.

**Preparation of cultured skin equivalents with or without trehalose**. The method used for LSE preparation was described previously[56]. Briefly, a collagen gel was prepared by mixing six volumes of ice-cold porcine collagen type I solution (Nitta Gelatin, Osaka, Japan) with one volume of 8 × DMEM, ten volumes of 1 × DMEM supplemented with 20% FBS, and one volume of 0.1 N NaOH, of which 1 ml was added to each culture insert (Transwell, 3-μm membrane pore, Corning, Corning, NY) in a six-well culture plate (Corning). Following polymerization of the gel in the inserts at 37 °C, two volumes of fibroblast suspension solution 5 × 10^5 cells/ml in 1 × DMEM supplemented with 10% FBS were added to eight volumes of the collagen solution (thus, the final collagen concentration was 0.8 mg/ml). Then, 3.5 ml of the fibroblast-containing collagen solution was applied to each insert. When the fibroblast-containing gel polymerized, DMEM supplemented with 10% FBS and ascorbic acid (final concentration 50 ng/ml) was added with or without trehalose (in three concentrations: 10, 30, and 100 mg/ml). The gel was submerged in culture for 5 days until the fibroblasts contracted the gel.

A larger LSE was constructed following the same method as previously described except using a larger culture insert (Transwell, 75-mm diameter, 3-μm membrane pore, Corning), thus utilizing proportionally more fibroblasts. A rubber ring (8-mm interior diameter) was covered over the fibroblast-containing gel to stabilize it within the large-scale LSE. In the hole of the ring, 6 × 10^5 keratinocytes in 30 μl MCDB 153 type II were seeded. The keratinocytes were submerged in culture for 2 days. When the keratinocytes reached confluence, the LSE was lifted to the air–liquid interface and a cornification medium (a 1:1 mixture of Ham's F-12 and DMEM supplemented with 2% FBS and other supplements, as described previously[56]) was added. The medium was changed every other day.

To construct the conventional LSE, keratinocytes were seeded onto the contracted gel and then submerged and airlifted as described above, except without the ring. The seeding cell density was adjusted using rubber rings. Both LSE types were harvested 7 or 14 days after airlifting. For hematoxylin and eosin staining, the LSE was fixed in 10% formalin and embedded in paraffin. For immunohistochemical staining, the LSE was snap-frozen in an OCT compound. We performed more than five experiments and obtained similar results. A representative experiment is depicted in Fig. 1. In comparative studies, keratinocytes and fibroblasts from the same donor were used.

**Transplanting cultured 3D dermal sheets**. The animal grafting protocol was approved by the Ethics Committee of Ehime University School of Medicine. Ten-week-old male BALB/cAJc1-nu nude mice (CLEA Japan, Tokyo, Japan) were anesthetized by isoflurane inhalation. Full-thickness wounds were created on the skin of the backs of each mouse using a 6-mm skin biopsy punch. The fibroblast-containing collagen gels were prepared with vehicle or trehalose (100 mg/ml) and submerged in culture for 5 days, and the dermal substitutes (1 day after airlift) were grafted onto the wounds, which were covered with transparent films: Mepitel One (Mölnlycke, Göteborg, Sweden) and Tegaderm (3 M Japan, Tokyo, Japan). Seven days after transplantation, the grafts were harvested. One part of each graft was paraffin-embedded and sectioned at 6 μm, from which hematoxylin and eosin staining was prepared. Some sections were de-paraffinized and blocked for endogenous peroxidase activity, and then blood vessels were stained with rat antibody against CD31 (at 1:20 dilution, dianova GmbH, Hamburg, Germany), according to the manufacturer's instructions for the ImmPRESS™ reagent kit (Vector Laboratories, Burlingame, CA). The sections were counterstained with hematoxylin for cell nuclei. We performed at least three independent studies and confirmed similar results. A representative experiment is shown in the figures.

**Whole transcriptome analysis with RNA-seq**. Total RNA was extracted from the fibroblasts or LSE using the RNeasy Mini Kit (Qiagen, Hilden, Germany), and mRNA was purified with oligo dT beads (NEBNext Poly (A) mRNA magnet Isolation Module, New England Biolabs, NEB, Ipswich, MA). The procedure of the complementary DNA (cDNA) libraries was performed with NEBNext Ultra II RNA library Prep kit (NEB) and NEBNextplex Oligos for Illumina following a previously described method[57]. Briefly, mRNA was incubated in NEBNext First Strand Synthesis Reaction Buffer at 94 °C for 15 min in the presence of NEBNext Random Primers, and reverse transcription was performed with NEBNext Strand Synthesis Enzyme Mix. The index sequences were inserted into the fragments with PCR amplification. The libraries were added in equal molecular amounts and were sequenced on an Illumina Next-seq DNA sequencer with a 75-bp pair-end cycle sequencing kit (Illumina, San Diego, CA). The detected reads were analyzed using CLC Genomics Workbench software (ver.8.01, Qiagen). The pathway for the detected genes was analyzed using IPA (Qiagen).

**Histology and immunohistochemical staining**. Paraffin-embedded LSE samples were sectioned at 6 μm and stained with hematoxylin and eosin (H&E) or alcian blue (pH 2.5). For immunohistochemical staining, the ImmPRESS™ reagent kit (Vector Laboratories) was used according to the manufacturer's instructions. Frozen sections (7 μm) were first incubated with 0.3% hydrogen peroxide for

30 min to remove endogenous peroxidase activity and then incubated overnight at 4 °C with primary antibodies at the appropriate dilutions. The antibodies used in this study were as follows: n1584 for α-SMA (Agilent Technologies, Santa Clara, CA) and NCL-Ki67-MM1 for Ki67 (LeicaBiosystems, Buffalo Grove, IL). The sections were incubated with enzyme-conjugated secondary antibodies for 30 min at room temperature, followed by the staining substrate. To determine if hyaluronan accumulates in LSEs, staining was conducted using a biotinylated hyaluronic acid-binding protein (Cosmo Bio). To detect elastic fibers in the tissue, EVG staining was performed using standard histological dyes (Muto Pure Chemicals, Tokyo, Japan). Images were obtained using a Nikon ECLIPSE E600 microscope coupled with Nikon DS-Ri1camera (Nikon, Tokyo, Japan).

**Evaluating the epidermal spreading potential of trehalose-treated LSEs**. Rubber rings with an inner diameter of 8 mm were put on gels with or without trehalose (10, 30, and 100 mg/ml) and $6 \times 10^5$ keratinocytes in 30 µl of MCDB 153 II medium were seeded within each ring. When the keratinocytes reached confluence, the LSEs were lifted to the air–liquid surface and the rubber rings were removed. At 14 days post-airlifting, the epidermal size was measured using computer-assisted morphometric analysis. The epidermal sizes of the conventional LSEs and trehalose-treated LSEs were statistically compared using Student's t-test.

**Cellular imaging using confocal laser-scanning fluorescence microscopy**. Fibroblasts were implanted from normal cell culture dishes to a glass-bottom culture plate (AGC TECHNO GLASS CO., LTD, Shizuoka, Japan) and incubated with trehalose (100 mg/ml) or vehicle for 24 h. For the exposure to oxidative stress, cells were exposed to hydrogen peroxide for 2 h, rinsed twice with phosphate-buffered saline (PBS), and incubated in DMEM with 1% FBS for 22 h before confocal imaging. The cells were examined with the Cellular Senescence Detection Kit-SPiDER-βGal to detect SA-βGal and DNA damage was measured using the DNA Damage Detection Kit (Dojindo Laboratories, Kumamoto, Japan) for confocal imaging. An A1R downlight laser-scanning confocal microscope (NIKON, Tokyo, Japan) was used for fixed-cell imaging. The excitation wavelengths were set to 405, 488, and 640 nm, respectively, and the emissions of the blue, green, and red channels were set to 425–475 nm, 500–530 nm, and over 662.5–737.5 nm, respectively. The laser illumination was set to 3.5, 20, and 20% power. Images were acquired using a Plan Apo λ 40× objective lens. Image stacks comprising more than ten optical sections with 1-µm Z-steps were acquired from the bottom of the glass, corresponding to an area of $317 \times 317$ µm$^2$ ($1024 \times 1024$ pixel$^2$, 0.31-µm pixel). The images were analyzed using the NIS element Ver 5.41 software package. A $3 \times 3$ median filter was applied to all images and indicated maximum intensity projection.

**p21 and dihydroethidium (DHE) immunocytochemistry**. For p21 immunocytochemistry, the treated fibroblasts were fixed in 4% paraformaldehyde/PBS for 30 min at room temperature, permeabilized with 0.5% Triton X-100/PBS for 15 min, incubated with antibodies raised against p21 #29475 (Cell Signaling Technology) overnight at 4 °C after blocking with blocking solution (S3022, Dako) for 30 min, and finally incubated with an Alexa Fluor 488-conjugated secondary antibody and DAPI (Thermo Fisher Scientific). The cells were mounted with VECTASHIELD (Vector Laboratories), and observed under a fluorescence microscope (Nikon), and the fluorescence data were analyzed by ImageJ (National Institutes of Health). For evaluating superoxide production, DHE staining (ab145360, Abcam, Cambridge, UK) was performed in the dark, and fluorescence images were captured using the fluorescence microscope BZ9000 (Keyence, Osaka, Japan) with the fluorescent filter OP-66838 (excitation 560/30 nm and emission 630/60 nm). Fluorescent signals were quantified using ImageJ (National Institutes of Health).

**Cell death assays**. Cell viability was measured using a Cell Counting Kit-8 assay (Enzo life sciences, Farmingdale, NY), following the manufacturer's instructions. Optical density was measured at 450 nm and normalized to the corresponding stimulation control.

**Cell cycle analysis of monolayer fibroblasts and dermal cells in LSEs**. Single-cell preparations from the monolayer fibroblasts or the dermal side of LSEs were performed. The dermis was removed from the epidermis of the LSE cells, digested with collagenase XI and hyaluronidase (Sigma-Aldrich) for 60 min at 37 °C, and analyzed by fluorescence-activated cell sorting (FACS) using propidium iodide (BioLegend, San Diego, CA), according to the propidium iodide cell cycle staining protocol. For cell cycle analysis, the BrdU Flow Kit (BD Biosciences, San Diego, CA) was used per the manufacturer's instructions. Cellular γH2AX was detected using the DNA Damage Detection Kit (Dojindo, Kumamoto, Japan), according to the manufacturer's instructions.

**RNA preparation and determination of mRNA expression by quantitative RT-PCR**. Total RNA was isolated using the RNeasy Mini Kit (Qiagen), and real-time PCR was used to determine the mRNA abundance, as described previously[58].

TaqMan™ Gene Expression Assays (Thermo Fisher Scientific) were used to analyze the gene expressions (Supplementary Table 2). GAPDH mRNA was used as an internal control. Target gene mRNA expression was calculated relative to GAPDH mRNA, and all data were normalized to their respective controls (mean of control cells or tissues).

**Small interfering RNA**. Silencer-validated siRNA CDKN1A (AM51331, Thermo Fisher Scientific) was used for silencing CDKN1A and the Silencer negative control siRNA (AM4611, Thermo Fisher Scientific) was used as control. Fibroblasts were transfected with siRNA using Lipofectamine RNAiMAX Transfection Reagent (Thermo Fisher Scientific), according to the manufacturer's instructions. The cells were allowed to stabilize for 24 h before trehalose or vehicle treatment.

**Western blotting analysis**. Following stimulation, total cell extracts were collected at the indicated times. To detect the protein levels, cell lysates were separated by SDS–polyacrylamide gel electrophoresis and transferred to polyvinylidene difluoride membranes. Analyses were performed using Amersham ECL Prime Western Blotting Detection Reagent (RPN2232) (GE Healthcare Life Sciences, Chicago, IL), and the membranes were scanned using Image Quant LAS4010 (GE Healthcare Life Sciences). We used primary antibodies for ERK (#9102), phospho-ERK (#4370), AKT (#9272), phospho-AKT (#9271), p21 waf/cipl (12p1) (#2947) (Cell Signaling Technology), Lamin B (C-20) (#sc-6216) (Santa Cruz, Dallas, TX), DPT (AF4629; R&D systems, Minneapolis, MN), and β-actin (#ab6276; Abcam).

**DPT ELISA**. DPT in the cell culture supernatants were measured. A Human DPT ELISA Kit (Abcam) was used to measure DPT in LSEs, according to the manufacturer's procedures.

**Statistics and reproducibility**. Statistical analysis was performed using a two-tailed Student's t-test, one- or two-way analysis of variance with Prism software (version 9; GraphPad Software, San Diego, CA). Results are expressed as mean ± standard deviation (SD). A P value of <0.05 was considered significant.

**Study approval**. This study was conducted according to the principles of the Declaration of Helsinki, and all procedures involving human subjects received previous approval from the ethics committee at Ehime University School of Medicine, Japan. All participants provided written informed consent. All animal procedures performed in this study were reviewed and approved by the Ehime University Institutional Animal Care and Use Committee. The experiments were conducted in accordance with the NIH guidelines for the care and use of animals and the recommendations of the International Association for the Study of Pain.

**Reporting summary**. Further information on research design is available in the Nature Portfolio Reporting Summary linked to this article.

## Data availability

All data were available in the paper or the supplementary materials. All numerical source data used for generating the main figures are uploaded as Supplementary Data. Uncropped and unedited blot images are included as Supplementary Blots. RNA sequence data were submitted to GEO under accession number GSE184892. The following data sets were generated. Jun Muto, Shinji Fukuda, Kenji Watanabe, Xiuju Dai, Teruko Tsuda, Takeshi Kiyoi, Hideki Mori, Ken Shiraishi, Masamoto Murakami, Shigeki Higashiyama, Yoichi Mizukami, Koji Sayama (2021) NCBI Gene Expression Omnibus ID GSE184892. Highly concentrated trehalose induces transient senescence-associated secretory phenotype in fibroblasts via CDKN1A/p21. https://www.ncbi.nlm.nih.gov/geo/query/acc.cgi?acc=GSE184892.

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

## Acknowledgements

This work was supported by TR SPRINT Stage-A Seeds (A209 and A054) from Japan Agency for Medical Research and Development (AMED) and JSPS KAKENHI Grant Number JP16H06280 and JP22H04926 Grant-in-Aid for Scientific Research on Innovative Areas—Platforms for Advanced Technologies and Research Resources "Advanced Bioimaging Support." The authors would like to thank E. Tan from Ehime University for performing the qPCR and flow cytometry experiments.

## Author contributions

J.M., S.F., K.W., X.D., T.T., T.K., K.K., R.K., H.M., K.S. (Ken Shiraishi), M.M., and Y.M. conducted experiments and/or contributed to the analysis of the results described in the paper. J.M., S.F., K.W., T.K., K.K., R.K., and Y.M. designed and analyzed the experiments. S.F., T.I., S.H., Y.F., Y.M., and K.S. (Koji Sayama) were involved in the supervision of the

work. J.M., S.F., and Y.M. wrote the manuscript with input from all the authors. J.M. obtained funding and has overall responsibility for the study.

## Competing interests

The authors declare no competing interests.
