## [Peer Review File · Communications Biology]

Reviewers' comments:

Reviewer #1 (Remarks to the Author):

In this work, Muto et al investigate the role of trehalose on living skin equivalent (LSE). They observed that high level of trehalose induces a high spread of LSE. Transcriptomic analyses of fibroblasts or keratinocytes with or without trehalose after long term culture show almost no DEGs induced by trehalose. By contrast short-term fibroblast treatment by trehalose provokes major transcriptomic changes. The data support that trehalose induces some cellular senescence and its SASP. SASP induction is dependent of p21. Graft of Trehalose-treated fibroblasts accelerated wound healing in vivo, eventually involving endothelial cells. Although interesting this works suffer from some weakness that need to be addressed (please see below).

I will also suggest to carefully improve the English as sometimes the text is a bit difficult to follow.

Comments:

- line 136-137 "Conversely, fibroblasts in the collagen gel with trehalose showed increased Ki67 positivity and proliferative capacity (Figure 1D)". Can the authors show a quantification to fully support the claim (not obvious from the images shown) ?

- line 202-205 "With respect to upstream factors, asparaginase, a drug for acute lymphoblastic leukemia, was detected in the upstream analysis and is able to arrest the cell cycle". I do not understand this sentence, asparaginase is not a drug.

. figure 2sup 1: I do not understand PC2 scale (inversion - and +) between C and F, can the authors explain ?

. from the figure 5D and E, I do not see evidence that trehalose-treated cells are arrested in G2/M phase. To support, this claim the authors need to do EdU staining and PI staining and if the cells are arrested in G2/M they should not integrate EdU but still be at 4n quantity of DNA.

. from figure 5E, I see more cells in S phase, when treated with trehalose. Again this has to be checked by EdU staining and comments in the text have to be adjusted. In addition, in Figure 1D, authors claim that trehalose increased proliferation according to Ki67, which sounds contradictory to their interpretation in figure 5E.

. mentioning G2/M interphase is not correct. G2 belongs to the interphase but M to mitosis. Please correct.

. Authors found increase pERK and pAKT after trehalose treatment. These phosphorylations can be seen as an oncogene stress (such as RAS-induced senescence) that is known to induce senescence by a replicative stress and subsequent DNA damage/p53/p21 pathway activation. Can the authors show whether they can detect some DNA damage after trehalose treatment?

. Authors should define the proportion of cells really senescing (% of p21 positive cells, % of SA-b-Gal positive cells).

. in Figure 7 the authors show that the SASP is dependent of p21, what about the other marks of senescence?

. In Figure 8 to support that the improvement in wound healing conferred by trehalose is linked to senescence, the authors need to repeat their assays with fibroblasts containing p21 siRNA ?

Reviewer #2 (Remarks to the Author):

This paper shows an interesting effect of high concentrations of a disaccharide of glucose that is not native to vertebrates (Trehalose) in inducing release of signalling molecules that promote wound healing. This appears to be regulated through p21 and shares some pathway similarity with senescence, albeit it in a transient manner.

This could have an important impact on the generation of skin constructs for, and in aiding wound repair through direct application. The increase in speed of generating living skin equivalents is striking and the promotion in wound healing seem very promising.

The experiments are well done, and I have no real technical comments about this.

However, extra clarity is needed in the time-lines of some experiments and I disagree with the interpretation of (or rather the wording of the interpretation) some of the results.

This mainly relates to differentiating a transient 'senescence-like' phenotype from irreversible 'traditional senescence'. As it currently stands I think this paper could just as easily be written as a pro-healing p21-signalling pathway response, without even mentioning senescence.

The authors suggest that their results show a transient (and with the inference that this is presumably reversible) induction of senescence, and a non-inflammatory and PDGFA enriched SASP that aids wound healing (possibly this is just how it was worded, as I mention later). This being different to the SASP induced by other stressors such as oncogene activation. In addition they also show changes in morphology, LaminB1 downregulation and Sen-B-Gal staining, and data surrounding Aurora-A kinase inhibition and a G2/M block.

I'm not clear on the timing of these experiments (Figure 5), as they state that cells are treated for 24 hours, but do not state how long after administration the stainings and effects are tested and observed at. Traditionally in senescence, the onset of senescence features such as morphological changes and Sen-B-Gal staining is delayed from the onset (often by up to and over 10 days). If these are (at least to what I could find in the text) occurring after only 24 hours in contact with Trehalose, then I would say that something else is probably occurring. A timeline, or stating how long after the seeming senescence induction these are tested for is required.

The existence of a transient and escapable senescence phenotype is a pretty bold claim - and I think would need a lot of evidence to back it up. While DiMaria has shown a transient induction of senescence in vivo, it is believed that these cells are removed by immune-cells rather than spontaneously reverting and doesn't occur in quite the same fashion in in vitro systems without immune-cell presence (in fairness increased CD31+ve cells are seen here).

If the authors actually meant that Trehalose induced a transient pro-repair SASP, and the cells remain senescent in vitro, but are cleared by immune cells in vivo - this could do with being a better explained. The current wording, and inclusion of sentences such as ("The role of CDKN1A is likely confined to the induction of senescence and cells can resume cycling upon resolution of stress (38)") suggest that they believe it is a transient event on a cellular level (i.e. transient pro-healing release of signalling molecules through p21 signalling). I also disagree with their interpretation of this reference as showing transient induction of senescence followed by resolution and escape.

The evidence I think does support that a pro-repair phenotype, that shares pathway similarities with senescence, is induced. I'm not yet convinced that it is 'true-senescence' due to the seeming transient nature of the phenotype (which almost by definition features irreversible cell-cycle arrest, as is not transient outside of escape in cancer). Quite possibly this is an existing biological mechanism that is also present & dysregulated in senescence.

The authors do later state 'we provide evidence that induction of the transient non-inflammatory senescence-like phenotype by trehalose in fibroblasts is beneficial for healing skin wounds' which I feel is a better interpretation. The abstract and final conclusion paragraph are also more restrained, and this should be reflected in text elsewhere. Or otherwise stress that this is 'senescence-like'. To state that it is senescence (rather than senescence-like or associated), would require a great deal further characterisation across a time-line, and tracking the proliferation of cells in culture across an extended period to see if there is a delayed reversal and an expansion of cells or not. I think either this needs to be investigated, or the authors take care to differentiate the transient events here from what is happening in 'traditional senescence'.

Otherwise, I enjoyed the paper and the authors have found something rather interesting.

Further notes:

It would have been nice to see a comparison with a stressor that does promote senescence, such as hydrogen peroxide and why Trehalose is probably a better idea.

The authors state 'Therefore, future senescence-targeted therapies with trehalose should be reserved for the treatment of chronic wounds of human diabetic patients', which seems at odds with the sentence in the next paragraph "This could provide the foundation of a new therapy to treat not only genodermatoses such as epidermolysis bullosa but also chronic diabetic and venous ulcers." This just needs tidying up or re-wording I think, unless they mean to only treat epidermolysis bullosa and venous ulcers in diabetic patients.

COMMSBIO-22-0730

Thank you for reviewing our manuscript and your thoughtful and constructive feedback. We are pleased to provide a revised manuscript that includes revisions responding to the reviewers' comments. We have revised the manuscript according to these comments.

Reviewer #1:

In this work, Muto et al investigate the role of trehalose on living skin equivalent (LSE). They observed that high level of trehalose induces a high spread of LSE. Transcriptomic analyses of fibroblasts or keratinocytes with or without trehalose after long term culture show almost no DEGs induced by trehalose. By contrast short-term fibroblast treatment by trehalose provokes major transcriptomic changes. The data support that trehalose induces some cellular senescence and its SASP. SASP induction is dependent of p21. Graft of Trehalose-treated fibroblasts accelerated wound healing in vivo, eventually involving endothelial cells. Although interesting this works suffer from some weakness that need to be addressed (please see below).

I will also suggest to carefully improve the English as sometimes the text is a bit difficult to follow.

Reply: Thank you for your suggestion. The manuscript has been edited and rewritten by an experienced scientific editor. I attached the certificate of editing.

Comments

-#1 line 136-137 “Conversely, fibroblasts in the collagen gel with trehalose showed increased Ki67 positivity and proliferative capacity (Figure 1D)”. Can the authors show a quantification to fully support the claim (not obvious from the images shown) ?

Reply: Thank you for your suggestion. We have added dotted lines and yellow arrow heads to indicate the dermal–epidermal junction and Ki67-positive fibroblasts on the dermal side of LSEs, respectively (Figure 1d). Furthermore, we have included a new figure quantifying the Ki67-positive fibroblasts, which shows that their number in the collagen gel with trehalose significantly increased (Supplementary Figure 2).

-#2 line 202-205 “With respect to upstream factors, asparaginase, a drug for acute lymphoblastic leukemia, was detected in the upstream analysis and is able to arrest the cell cycle”. I do not understand this sentence, asparaginase is not a drug.

Reply: Asparaginase is originally classified as an “enzyme” and currently used as an antineoplastic chemotherapy drug (Elspar or Leunase) to treat acute lymphocytic leukemia.

-#3 figure 2sup 1: I do not understand PC2 scale (inversion – and +) between C and F, can the authors explain ?

Reply: Previous Figure 2-figure supplement1 C and F (Supplementary Figure 5c, f) show the factor loadings of the genes contributing to the separation of previous Figure 2-figure supplement1 B and E (Supplementary Figure 5b, e), respectively. In the 2D fibroblasts of previous Figure 2-figure supplement1B (Supplementary Figure 5b), trehalose shifted toward the positive direction of PC2, showing that the genes contributing to trehalose separation were present in the positive region. In the 3D fibroblasts of previous Figure 2-figure supplement1F (Supplementary Figure 5f), trehalose moved toward the negative direction of PC2, indicating that genes contributing to trehalose separation existed in the negative region. To display the genes that contributed to trehalose separation in the same direction, the PC2 axis in previous Figure 2-figure supplement1F (Supplementary Figure 5f) was inverted to that seen in previous Figure 2-figure supplement1C (Supplementary Figure 5c).

-#4 from the figure 5D and E, I do not see evidence that trehalose-treated cells are arrested in G2/M phase. To support, this claim the authors need to do EdU staining and PI staining and if the cells are arrested in G2/M they should not integrate EdU but still be at 4n quantity of DNA.

Reply: Thank you very much for this valuable suggestion. We have performed BrdU staining and flow cytometric analysis with a dye binding to total DNA (7-AAD). We confirmed that following treatment with highly concentrated trehalose, significantly more number of 2D fibroblasts were actually arrested in the G2/M phase and the number of cells in the S phase decreased significantly within 24 hours, as shown in the new figures (Figure 5g, h). We have revised the text to clarify these points (see lines 257–261).

-#5 from figure 5E, I see more cells in S phase, when treated with trehalose. Again this has to be checked by EdU staining and comments in the text have to be adjusted.

Reply: Thank you for this comment. Interestingly, 32% of the 3D fibroblasts cultivated with trehalose (100 mg/ml) were detected in the S and G2/M phases after 7 days, whereas almost none of the untreated cells were detected in these phases in the previous Figure 5E (new figure 5i). In addition, in new Figure 5i, we found that trehalose sustained the proliferation although the vehicle-treated control fibroblasts in 3D culture were arrested in the G1 phase after 1 week. We have modified the text accordingly (See lines 263–267).

Furthermore, we performed staining of incorporated BrdU and flow cytometric analysis with 7-AAD on single cell suspensions of the 3D culture of fibroblasts after 1 week. Unfortunately, we could not get stable data that could be included in the manuscript due to technical difficulties in recovering the cells from the collagen gel for flow cytometric analysis.

-In addition, in Figure 1D, authors claim that trehalose increased proliferation according to Ki67, which sounds contradictory to their interpretation in figure 5E.

Reply: Thank you very much for raising an important question. We found that high-concentration trehalose treatment induced a senescence-like state in fibroblasts, upregulating several growth factors, such as FGF2, VEGFA, EREG, and ANGPT2, in the early phase (between 24–72 hours). These growth factors, secreted extracellularly, may sustain the proliferation of the fibroblasts in LSE during the later phase (after 2 weeks), as seen in Figure 1d.

-#6 mentioning G2/M interphase is not correct. G2 belongs to the interphase but M to mitosis. Please correct.

Reply: Thank you for pointing this out and we apologize for this error. We have corrected it to G2/M-phase.

-#7 Authors found increase pERK and pAKT after trehalose treatment. These phosphorylations can be seen as an oncogene stress (such as RAS-induced senescence) that is known to induce senescence by a replicative stress and subsequent DNA damage/p53/p21 pathway activation. Can the authors show whether they can detect some DNA damage after trehalose treatment?

Reply: Thank you very much for these comments. We used the DNA Damage Detection Kit to monitor H2AX with phosphorylated Ser-139 (γ H2AX) as a DNA damage marker. As described in the manuscript (lines 242–251), we observed that both high-concentration trehalose and vehicle treatment of the fibroblasts induced similar DNA damage but hydrogen peroxide treatment induced remarkably more damage than trehalose, as evidenced by monitoring H2AX with phosphorylated Ser-139 (γ H2AX) using confocal microscopy (Figure 5b). Moreover, we performed flow cytometric analysis revealed that the percentage of DNA-damaged cells non-significantly differed between the high-concentration trehalose and vehicle-treated groups. However, it significantly increased after hydrogen peroxide treatment, as seen by monitoring γ H2AX (Figure 5c, d). These findings indicate that high-concentration trehalose induces a senescence-like state with dramatically less DNA damage than hydrogen peroxide.

-#8 Authors should define the proportion of cells really senescing (% of p21 positive cells, % of SA-b-Gal positive cells).

Reply: Thank you for this suggestion. We have differentiated “traditional irreversible senescence” from a transient “senescence-like” phenotype and revised the title and manuscript by using this key term, “senescence-like.” We believe that highly concentrated trehalose did not induce senescence but rather a senescence-like state, as described in the manuscript. Therefore, we think that in discussing truly senescent cells, we would be departing from the main point of this paper.

We have also showed the proportion of p21-positive cells and p21-positive intensity/field in Supplementary Figure 7. Furthermore, we used the Cellular Senescence Detection Kit-SPiDER-βGal to detect SA-βGal-positive cells and found that markedly more trehalose-treated fibroblasts (87%) were SA-βGAL-positive than vehicle-treated fibroblasts (14%) (Figure 5b).

-#9 in Figure 7 the authors show that the SASP is dependent of p21, what about the other marks of senescence?

Reply: Thank you very much for asking this important question. We have differentiated a transient “senescence-like” phenotype from “traditional irreversible senescence” and revised the paper accordingly. The key term, “senescence-like,” has been used throughout the manuscript. We showed that highly concentrated trehalose transiently induced a “non-inflammatory, growth factor-enriched, p21-dependent pro-wound healing state,” which differs from the SASP induced by other stress factors. We agree that investigating the involvement of p21 in the highly concentrated trehalose-induced senescence-like state is important, and we are actively pursuing this in our lab using RNAseq analysis and other techniques. Therefore, the characterization of both trehalose and CDKN1A siRNA-treated fibroblasts would fall outside the scope of this study.

-#10 In Figure 8 to support that the improvement in wound healing conferred by trehalose is linked to senescence, the authors need to repeat their assays with fibroblasts containing p21 siRNA ?

Reply: Thank you very much for this suggestion. We have performed *in vivo* experiments using a dermal sheet containing CDKN1A siRNA-treated fibroblasts and have added the new results to the manuscript (Figure 9, lines 334–351). These results indicate that highly concentrated trehalose induced a senescence-like state and promoted wound healing via the p21 signaling pathway.

Reviewer #2 (Remarks to the Author):

This paper shows an interesting effect of high concentrations of a disaccharide of glucose that is not native to vertebrates (Trehalose) in inducing release of signalling molecules that promote wound healing. This appears to be regulated through p21 and shares some pathway similarity with senescence, albeit it in a transient manner.

This could have an important impact on the generation of skin constructs for, and in aiding wound repair through direct application. The increase in speed of generating living skin equivalents is striking and the promotion in wound healing seem very promising.

The experiments are well done, and I have no real technical comments about this.

However, extra clarity is needed in the time-lines of some experiments and I disagree with the interpretation of (or rather the wording of the interpretation) some of the results.

This mainly relates to differentiating a transient 'senescence-like' phenotype from irreversible 'traditional senescence'. As it currently stands I think this paper could just as easily be written as a pro-healing p21-signalling pathway response, without even mentioning senescence.

The authors suggest that their results show a transient (and with the inference that this is presumably reversible) induction of senescence, and a non-inflammatory and PDGFA enriched SASP that aids wound healing (possibly this is just how it was worded, as I mention later). This being different to the SASP induced by other stressors such as oncogene activation. In addition they also show changes in morphology, LaminB1 downregulation and Sen-B-Gal staining, and data surrounding Aurora-A kinase inhibition and a G2/M block.

I'm not clear on the timing of these experiments (Figure 5), as they state that cells are treated for 24 hours, but do not state how long after administration the stainings and effects are tested and observed at. Traditionally in senescence, the onset of senescence features such as morphological changes and Sen-B-Gal staining is delayed from the onset (often by up to and over 10 days). If these are (at least to what I could find in the text) occurring after only 24 hours in contact with Trehalose, then I would say that something else is probably occurring. A timeline, or stating how long after the seeming senescence induction these are tested for is required.

Reply:

Thank you very much for your comments. We treated 2D fibroblasts with high-concentration trehalose for 24 or 48 h (Figure 5a, b, c, d, e, f, g, h) as the early phases, while 3D fibroblasts were treated for 72 h as the early phase (Figure 4, 6). Immediately after incubating with trehalose or vehicle, we harvested the cells and tested for the induction of a senescence-like state in fibroblasts. Furthermore, we harvested the samples of final LSE preparations 2 weeks after trehalose

treatment (late phase) for RNAseq and pathological analysis. We have added the incubation times and timing of the tests in the manuscript and figure legends.

The existence of a transient and escapable senescence phenotype is a pretty bold claim - and I think would need a lot of evidence to back it up. While DiMaria has shown a transient induction of senescence in vivo, it is believed that these cells are removed by immune-cells rather than spontaneously reverting and doesn't occur in quite the same fashion in in vitro systems without immune-cell presence (in fairness increased CD31+ve cells are seen here).

If the authors actually meant that Trehalose induced a transient pro-repair SASP, and the cells remain senescent in vitro, but are cleared by immune cells in vivo - this could do with being a better explained. The current wording, and inclusion of sentences such as ("The role of CDKN1A is likely confined to the induction of senescence and cells can resume cycling upon resolution of stress (38)") suggest that they believe it is a transient event on a cellular level (i.e. transient pro-healing release of signalling molecules through p21 signalling). I also disagree with their interpretation of this reference as showing transient induction of senescence followed by resolution and escape.

The evidence I think does support that a pro-repair phenotype, that shares pathway similarities with senescence, is induced. I'm not yet convinced that it is 'true-senescence' due to the seeming transient nature of the phenotype (which almost by definition features irreversible cell-cycle arrest, as is not transient outside of escape in cancer). Quite possibly this is an existing biological mechanism that is also present & dysregulated in senescence.

The authors do later state 'we provide evidence that induction of the transient non-inflammatory senescence-like phenotype by trehalose in fibroblasts is beneficial for healing skin wounds' which I feel is a better interpretation. The abstract and final conclusion paragraph are also more restrained, and this should be reflected in text elsewhere. Or otherwise stress that this is 'senescence-like'. To state that it is senescence (rather than senescence-like or associated), would require a great deal further characterisation across a time-line, and tracking the proliferation of cells in culture across an extended period to see if there is a delayed reversal and an expansion of cells or not. I think either this needs to be investigated, or the authors take care to differentiate the transient events here from what is happening in 'traditional senescence'.

Reply:

We would like to thank the reviewer for these thoughtful comments and insights. We agree with you and have integrated this suggestion throughout our manuscript to clarify these points. We have stressed that this is “senescence-like”, and differentiated the transient novel event from “traditional senescence” to align with your comments.

Otherwise, I enjoyed the paper and the authors have found something rather interesting.

Reply:
Thank you very much for your kind words.

Further notes:

It would have been nice to see a comparison with a stressor that does promote senescence, such as hydrogen peroxide and why Trehalose is probably a better idea.

Reply:
Thank you for raising an important point. We used low concentration hydrogen peroxide as a stressor, and found that hydrogen peroxide significantly induced more DNA damage in fibroblasts than highly concentrated trehalose (Figure 5b, c, and d), demonstrating that highly concentrated trehalose treatment might be a more beneficial method to induce the pro-healing p21 signaling pathway response without inducing DNA damage.

The authors state 'Therefore, future senescence-targeted therapies with trehalose should be reserved for the treatment of chronic wounds of human diabetic patients', which seems at odds with the sentence in the next paragraph 'This could provide the foundation of a new therapy to treat not only genodermatoses such as epidermolysis bullosa but also chronic diabetic and venous ulcers.' This just needs tidying up or re-wording I think, unless they mean to only treat epidermolysis bullosa and venous ulcers in diabetic patients.

Reply:
We agree with you and have corrected the sentence (lines 486–488, 494-495).

REVIEWERS' COMMENTS:

Reviewer #1 (Remarks to the Author):

The authors addressed my comments. In my opinion the article is now suitable for publication.

Reviewer #2 (Remarks to the Author):

I thank the authors for their work and corrections.

I don't think further changes are required to address the points I raised, and the changes add clarity to the experiments and interpretation.